# Uncoupled and Convergent Learning in Monotone Games under Bandit Feedback

## Abstract

We study the problem of no-regret learning algorithms for general monotone and smooth games and their last-iterate convergence properties. Specifically, we investigate the problem under bandit feedback and strongly uncoupled dynamics, which allows modular development of the multi-player system that applies to a wide range of real applications. We propose a mirror-descent-based algorithm, which converges in $O(T^{-1/4})$ and is also no-regret. The result is achieved by a dedicated use of two regularizations and the analysis of the fixed point thereof. The convergence rate is further improved to $O(T^{-1/2})$ in the case of strongly monotone games. Motivated by practical tasks where the game evolves over time, the algorithm is extended to time-varying monotone games. We provide the first non-asymptotic result in converging monotone games and give improved results for equilibrium tracking games.

## 1 Introduction

We consider multi-player online learning in games. In this problem, the cost function for each player is unknown to the player, and they need to learn to play the game through repeated interaction with other players. We focus on a class of monotone and smooth games, which was first introduced by Rosen (1965). This encapsulates a wide array of common games, such as two-player zero-sum games, convex-concave games, and zero-sum polymatrix games (Bregman & Fokin, 1987). Our goal is to find algorithms that solve the problem under bandit feedback and strongly uncoupled dynamics. Within this context, each player can only access information regarding the cost function associated with their chosen actions without prior insight into their counterparts. This allows modular development of the multi-player system in real applications and leverages existing single-agent learning algorithms for reuse.

Many works have focused on the time-average convergence to Nash equilibrium on learning in monotone games (Even-Dar et al., 2009; Syrgkanis et al., 2015; Farina et al., 2022). However, these works only guarantee the convergence of the time average of the joint action profile. Such convergence properties are less appealing, because while the trajectories of the players converge in the time-average sense, it may still exhibit cycling (Mertikopoulos et al., 2018). This jeopardizes the practical use of such algorithms.

Popular no-regret algorithms such as mirror descent have demonstrated convergence in the last iterate within specific scenarios, such as two-player zero-sum games (Cai et al., 2023) and strongly monotone games (Bravo et al., 2018; Drusvyatskiy et al., 2022; Lin et al., 2021). Yet convergence to Nash equilibrium in monotone and smooth games is not available unless one assumes exact gradient feedback and coordination of players (Cai et al., 2022; Cai & Zheng, 2023). It remains open as to whether a no-regret algorithm can efficiently converge to a Nash equilibrium in monotone games with bandit feedback and strongly uncoupled dynamics. In this paper, we investigate the pivotal question:

*How fast can no-regret algorithms converge (in the last iterate) to a Nash equilibrium in general monotone and smooth games with bandit feedback and strongly uncoupled dynamics?*

In this work, we present a mirror-descent-based algorithm designed to converge to the Nash equilibrium in monotone and smooth games. Our algorithm is uncoupled and convergent and is applicable

to the general monotone and smooth game setting. Motivated by real applications, where many games are also time-varying, we extend our study to encompass time-varying monotone games. This justifies that our algorithm could be deployed in both stationary and non-stationary tasks.

We achieve state-of-the-art results in both monotone games and time-varying monotone games.

- In monotone and smooth games:
    - Under bandit feedback and strongly uncoupled dynamics, we show our algorithm achieves a last-iterate convergence rate of $O(T^{-1/4})$.
    - In cases where the game exhibits strong monotonicity, our result improves to $O(T^{-1/2})$, matching the current best available convergence rates for strongly monotone games (Drusvyatskiy et al., 2022; Lin et al., 2021).
    - Our algorithm is no regret albeit players may be self-interested. The individual regret is at most $O(T^{3/4})$ in monotone games and at most $O(T^{1/2})$ in strongly monotone games.
- In time-varying monotone and smooth games:
    - If the game eventually converges to a static state within a time frame of $O(T^{\alpha})$, our algorithm achieves convergence in $O(T^{-1/4+\alpha})$.
    - If the game does not converge but experiences gradual changes in the Nash equilibrium that evolves in $O(T^{\varphi})$, our algorithm exhibits convergence rates of $O\left(\max\left\{T^{2\varphi/3-2/3}, T^{(4\varphi+5)^2/72-9/8}\right\}\right)$. The algorithm outperforms best available results of $T^{\varphi/5-1/5}$ by Duvocelle et al. (2023) and $T^{\varphi/3-2/3}$ by Yan et al. (2023).

Table 1 and Table 2 summarize our results and the results of previous works.

## 2 RELATED WORKS

**Monotone games** The convergence of monotone games has been studied in a significant line of research. For a strongly monotone game under exact gradient feedback, the linear last-iterate convergence rate is known (Tseng, 1995; Liang & Stokes, 2019; Zhou et al., 2020). Under noisy gradient feedback, Jordan et al. (2023) showed a last-iterate convergence rate of $O(T^{-1})$. Under bandit feedback, Bervoets et al. (2020) proposed an algorithm that asymptotically converges to the equilibrium if it is unique. Bravo et al. (2018) subsequently introduced an algorithm with a last-iterate convergence rate of $O(T^{-1/3})$, while also ensuring the no-regret property. Later works (Lin et al., 2021) further improved the last-iterate convergence rate to $O(T^{-1/2})$ under bandit feedback using the self-concordant barrier function. Jordan et al. (2023) gave a result of the same rate, but with the additional assumption that the Jacobian of each player's gradient is Lipschitz continuous. In the case of bandit but noisy feedback (with a zero-mean noise), Lin et al. (2021) showed that the convergence rate is still $O(T^{-1/2})$.

For monotone but not strongly monotone games, Mertikopoulos & Zhou (2019) leveraged the dual averaging algorithm to demonstrate an asymptotic convergence rate under noisy gradient feedback. With access to the exact gradient information, Cai & Zheng (2023) gave a last-iterate convergence rate of $O(T^{-1})$. In the context of bandit feedback, Tatarenko & Kamgarpour (2019) proposed an algorithm that asymptotically converges to the Nash equilibrium. Table 1 provides a summary of the recent results.

**Time-varying monotone games** Motivated by real-world applications such as Cournot competition, where multiple firms supply goods to the market and pricing is subject to fluctuations due to factors like weather, holidays, and politics. Duvocelle et al. (2023) studied the strongly monotone game under a time-varying cost function. When the game converges to a static state, they propose an algorithm that achieves asymptotic convergence under bandit feedback. Assuming the cost function varies $O(T^{\phi})$ across a horizon $T$, Duvocelle et al. (2023) provided an algorithm that attains a convergence rate of $O(T^{\phi/5-1/5})$ under bandit feedback. Subsequent work of Yan et al. (2023) further improved this rate to $O(T^{\phi/3-2/3})$ under exact gradient feedback.

Table 1: Summary of results for monotone games. "E" stands for the result in expectation and "P" stands for the result held in high probability. Strongly monotone games are abbreviated to "StroM", while monotone games are abbreviated to "M". We use "linear*" to denote the two-player zero-sum game, which is a special case of the linear game. We use "(N)" to remark that the results can also be obtained with noisy feedback.

| | Class of games | Feedback | Results |
|---|---|---|---|
| Bravo et al. (2018) | StroM | bandit | $O(T^{-1/3})$ (E) |
| Drusvyatskiy et al. (2022) | StroM | bandit | $O(T^{-1/2})$ (E) |
| Lin et al. (2021) | StroM | bandit (N) | $O(T^{-1/2})$ (E) |
| Jordan et al. (2023) | StroM | noisy gradient | $O(T^{-1})$ |
| **Ours** | StroM | bandit (N) | $O(T^{-1/2})$ (E & P) |
| Mertikopoulos & Zhou (2019) | M | noisy gradient | asymptotic |
| Cai & Zheng (2023) | M | exact gradient | $O(T^{-1})$ |
| Tatarenko & Kamgarpour (2019) | M | bandit | asymptotic |
| **Ours** | M | bandit (N) | $O(T^{-1/4})$ (E) |
| Cai et al. (2023) | linear* | bandit | $O(T^{-1/6})$ (E) |
| **Ours** | linear | bandit | $O(T^{-1/6})$ (E) |

Table 2: Summary of last-iterate convergence results for time-varying games. All results here are in expectation results. Strongly monotone games are abbreviated to "StroM", and monotone games are abbreviated to "M".

| | Class of games | Time-varying property | Feedback | Results |
|---|---|---|---|---|
| Duvocelle et al. (2023) | StroM | converging in $O(T^\alpha)$ | bandit | asymptotic |
| **Ours** | M | converging in $O(T^\alpha)$ | bandit | $O(T^{-1/4+\alpha})$ |
| Duvocelle et al. (2023) | StroM | $O(T^\varphi)$ variation path | bandit | $O(T^{\varphi/5-1/5})$ |
| Yan et al. (2023) | StroM | $O(T^\varphi)$ variation path | exact gradient | $O(T^{\varphi/3-2/3})$ |
| **Ours** | M | $O(T^\varphi)$ variation path | bandit | $O\left(\max\{T^{2\varphi/3-2/3}, T^{(4\varphi+5)^2/72-9/8}\}\right)$ |

## 3 PRELIMINARIES

We consider a multi-player game with $n$ players, with the set of players denoted as $\mathcal{N}$. Each player $i$ takes action on a compact and convex set $\mathcal{X}_i \subseteq \mathbb{R}^d$ of $d$ dimensions, and has cost function $c_i(x_i, x_{-i})$, where $x_i \in \mathcal{X}_i$ is the action of the $i$-th player and $x_{-i} \in \prod_{j \in [n], j \neq i} \mathcal{X}_j$ is the action of all other players. We assume the radius of $\mathcal{X}_i$ is bounded, i.e., $\|x - x'\| \leq B, \forall x, x' \in \mathcal{X}_i$. Without loss of generality, we further assume $c_i(x) \in [0, 1]$.

In this work, we study a class of monotone continuous games, where the gradient of the cost functions is monotone and the cost functions continuous (Assumption 3.1). Games that satisfy this assumption include convex-concave games, convex potential games, extensive form games, Cournot competition, and splittable routing games. A discussion of these games is available in Section 3.1. Note that the class of monotone continuous games is commonly studied in the literature (Lin et al., 2021; Farina et al., 2022).

**Assumption 3.1.** *For all player $i \in \mathcal{N}$, the cost function $c_i(x_i, x_{-i})$ is continuous, differentiable, convex, and $\ell_i$-smooth in $x_i$. Further, $c_i$ has bounded gradient $|\nabla_i c_i(x)| \leq G$ and the gradient $F(x) = [\nabla_i c_i(x)]_{i \in \mathcal{N}}$ is a monotone operator, i.e., $(F(x) - F(y))^\top (x - y) \geq 0, \forall x, y$.*

For notational convenience, we denote $L = \sum_{i \in \mathcal{N}} \ell_i$.

A common solution concept in the game is Nash equilibrium, which is a state of dynamic where no player can reduce its cost by unilaterally changing its action. Our aim is to learn a Nash equilibrium $x^* \in \prod_i \mathcal{X}_i$ of the game. Formally, the Nash equilibrium is defined as follows.

**Definition 3.1** (Nash equilibrium). *An action* $x^* \in \prod_i \mathcal{X}_i$ *is a Nash equilibrium if* $c_i(x^*) \leq c_i(x_i, x^*_{-i})$, $\forall x_i \in \mathcal{X}_i, x_i \neq x^*_i, i \in \mathcal{N}$.

When the game satisfies Assumption 3.1, and is with a compact action set, it is known that it must admit at least one Nash equilibrium (Debreu, 1952).

## 3.1 Examples of Monotone Continuous Games

A wide range of monotone games are captured by Assumption 3.1, and we now present a few classic examples. We include more examples in the appendix.

**Example 3.1** (convex-concave game). *Consider a two-player convex-concave game, where the objective function is* $c_1(x_1, x_2) = f(x_1, x_2)$, $c_2(x_1, x_2) = -f(x_1, x_2)$. *It is immediate that if* $f$ *is continuous, differentiable, smooth, convex in* $x_1$, *concave in* $x_2$, *then the game satisfies Assumption 3.1. Examples are rock paper scissors and chicken games.*

**Example 3.2** (Cournot competition). *In the Cournot oligopoly model, there is a finite set of* $N$ *firms, where firm* $i$ *supplies the market with a quantity* $x_i \in [0, C_i]$ *of some good and* $C_i$ *is the firm's production capacity. The good is priced as a decreasing function* $P(x_{\text{tot}}) = a - b x_{\text{tot}}$, *where* $x_{\text{tot}} = \sum_{i=1}^{N} x_i$ *is the total number of goods supplied to the market, and* $a, b > 0$ *are positive constants. The cost of firm* $i$ *is then given by* $c_i(x_i, x_{-i}) = d_i x_i - x_i P(x_{\text{tot}})$, *where* $d_i$ *is the cost of producing one unit of good. This is the associated production cost minus the total revenue from producing* $x_i$ *units of goods. It is clear that* $c_i$ *is continuous and differentiable, and Bravo et al. (2018) showed* $c_i$ *has positive definite and bounded hessian (is convex and smooth).*

**Example 3.3** (Splittable routing game). *In a splittable routing game, each player directs a flow, denoted as* $f_i$, *from a source to a destination within an undirected graph* $G = (V, E)$. *Each edge* $e \in E$ *is linked to a latency function, represented as* $\ell_e(f)$, *which denotes the latency cost of the flow passing through the edge. The strategies available to player* $i$ *are the various ways of dividing or "splitting" the flow* $f_i$ *into distinct paths connecting the source and the destination. With some restrictions on the latency function, the game satisfies Assumption 3.1 (Roughgarden & Schoppmann, 2015).*

## 3.2 Bandit Feedback and Strongly Uncoupled Dynamic

In this work, we focus on learning under bandit feedback and strongly uncoupled dynamics. The bandit feedback setting restricts each player to only observe the cost function $c_i(x_i, x_{-i})$ with respect to the action taken $x_i$. The strongly uncoupled learning dynamic (Daskalakis et al., 2011) means players do not have prior knowledge of cost function or the action space of other players and can only keep track of a constant amount of historical information. As the bandit feedback and strongly uncoupled dynamic only require each player to access information of its own, this allows for modular development of the multi-player system, by reusing existing single-agent learning algorithms.

# 4 Algorithm

Our algorithm builds upon the renowned mirror-descent algorithm. The efficacy of online mirror-descent in solving Nash equilibrium has been demonstrated under full information, and in both linear or strongly monotone games, with extensive investigations into its last-iterate convergence investigated in Cen et al. (2021); Lin et al. (2021); Cai et al. (2023); Duvocelle et al. (2023).

Our algorithm differs from classic online mirror descent approaches by making use of two regularizers: A self-concordant barrier regularizer $h$ to build an efficient Ellipsoidal gradient estimator and contest the bandit feedback; and a regularizer $p$ to accommodate monotone (and not strongly monotone) games. Similar use of two regularizers has also been investigated (Lin et al., 2021). However, their method used the Euclidean norm regularization, which cannot be extended to our setting.

**Regularizers** Let $h$ be a $\nu$-self-concordant barrier function (Definition 4.1), $p$ be a convex function with $\mu I \preceq \nabla^2 p(x) \preceq \zeta I$, $\zeta > 0, \mu \geq 0$. Let $D_p$ denote the Bregman divergence induced by $p$. We choose $p$ such that for any $x_i, x_i' \in \mathcal{X}_i$, $D_p(x_i, x_i') \leq C_p < \infty$, and for some $\kappa > 0$, $c_i(x_i, x_{-i}) - \kappa p(x_i)$ to be convex. Notice that when $c_i$ is convex but not linear, we can always find such $p$ when the action set is bounded. Intuitively, this is to interpolate a function $p$ that possesses less curvature than all $c_i$. We will discuss the modification to the algorithm needed when $c_i$ is linear in Section 5.3.

**Definition 4.1.** *A function $h : int(\mathcal{X}) \mapsto \mathbb{R}$ is a $\nu$-self concordant barrier for a closed convex set $\mathcal{X} \subseteq \mathbb{R}^n$, where $int(\mathcal{X})$ is an interior of $\mathcal{X}$, if 1) $h$ is three times continuously differentiable; 2) $h(x) \to \infty$ if $x \to \partial\mathcal{X}$, where $\partial\mathcal{X}$ is a boundary of $\mathcal{X}$; 3) for $\forall x \in \mathrm{int}(\mathcal{X})$ and $\forall \lambda \in \mathbb{R}^n$, we have $\left|\nabla^3 h(x)[\lambda, \lambda, \lambda]\right| \leq 2\left(\lambda^\top \nabla^2 h(x)\lambda\right)^{3/2}$ and $\left|\nabla h(x)^\top \lambda\right| \leq \sqrt{\nu}\left(\lambda^\top \nabla^2 h(x)\lambda\right)^{1/2}$ where $\nabla^3 h(x)[\lambda_1, \lambda_2, \lambda_3] = \frac{\partial^3}{\partial t_1 \partial t_2 \partial t_3} h\left(x + t_1\lambda_1 + t_2\lambda_2 + t_3\lambda_3\right)\Big|_{t_1=t_2=t_3=0}$.*

1. *$h$ is three times continuously differentiable;*

2. *$h(x) \to \infty$ if $x \to \partial\mathcal{X}$, where $\partial\mathcal{X}$ is a boundary of $\mathcal{X}$;*

3. *for $\forall x \in \mathrm{int}(\mathcal{X})$ and $\forall \lambda \in \mathbb{R}^n$, we have $\left|\nabla^3 h(x)[\lambda, \lambda, \lambda]\right| \leq 2\left(\lambda^\top \nabla^2 h(x)\lambda\right)^{3/2}$ and $\left|\nabla h(x)^\top \lambda\right| \leq \sqrt{\nu}\left(\lambda^\top \nabla^2 h(x)\lambda\right)^{1/2}$ where $\nabla^3 h(x)[\lambda_1, \lambda_2, \lambda_3] = \frac{\partial^3}{\partial t_1 \partial t_2 \partial t_3} h\left(x + t_1\lambda_1 + t_2\lambda_2 + t_3\lambda_3\right)\Big|_{t_1=t_2=t_3=0}$.*

It is shown that any closed convex domain of $\mathbb{R}^d$ has a self-concordant barrier (Lee & Yue, 2021).

**Ellipsoidal gradient estimator** As our algorithm operates under bandit feedback and strongly uncoupled dynamics, we would need to design a gradient estimator while only using costs for the individual player.

Let $\mathbb{S}^d$, $\mathbb{B}^d$ be the $d$-dimensional unit sphere and the $d$-dimensional unit ball, respectively. Our algorithm estimates the gradient using the following ellipsoidal estimator:

$$\hat{g}_i^t = \frac{d}{\delta_t} c_i(\hat{x}_i^t)(A_i^t)^{-1} z_i^t, \quad A_i^t = (\nabla^2 h(x_i^t) + \eta_t(t+1)\nabla^2 p(x_i^t))^{-1/2}, \quad \hat{x}_i^t = x_i^t + \delta_t A_i^t z_i^t,$$

where $z_i^t$ is uniformly independently sampled from $\mathbb{S}^d$ and $\delta_t, \eta_t \in [0, 1]$ are tunable parameters.

One can show that $\hat{g}_i^t$ is an unbiased estimate of the gradient of a smoothed cost function $\hat{c}_i(x^t) = \mathbb{E}_{w_i^t \sim \mathbb{B}^d} \mathbb{E}_{\mathbf{z}_{-i}^t \sim \Pi_{j \neq i} \mathbb{S}^d} \left[c_i\left(x_i^t + A_i^t w_i^t, \hat{x}_{-i}^t\right)\right]$. When $p$ is strongly convex, one can upper bound $\|\nabla_i \hat{c}_i(x) - \nabla_i c_i(x)\|$ by the maximum eigenvalue of $A_i^t$ and it suffices to take $\delta_t = 1$, which recovers the results in Lin et al. (2021). However, when $p$ is convex and not strongly convex, one would need to carefully tune $\delta_t$ to control the bias from estimating the smoothed cost function. This ellipsoidal gradient estimator was first introduced by Abernethy et al. (2008) for the case of $c_i$ being linear, and was then extended by Hazan & Levy (2014) to the case of strongly convex costs. In learning for games, the ellipsoidal estimator was used in the case of strongly monotone games (Bravo et al., 2018; Lin et al., 2021).

Based on the ellipsoidal gradient estimator, we present our uncoupled and convergent algorithm for monotone games under bandit feedback.

**Implementation** Notice that solving Equation (1) is equivalent to solving a convex but potentially non-smooth optimization problem. Certain sets $\mathcal{X} \subseteq \mathbb{R}^d$, including the cases when $\mathcal{X}$ is the strategy space of a normal-form game or an extensive-form game, can be solved by proximal Newton algorithm provably in $O(\log^2(1/\epsilon))$ iterations (Farina et al., 2022). When such guarantees are not required, one could accommodate other optimization methods in solving (1). Our experiment section provides more details.

The choice of $p$ and $h$ is game-dependent. For example, when $c_i(x) = x^2$ and the action set is on the positive half line, we can use the negative log function as our self-concordant barrier function $h$ and take $p = x$.

---

**Algorithm 1:** Algorithm

**Input:** Learning rate $\eta_t$, parameter $\delta_t$, regularizer $h(\cdot), p(\cdot)$, constant $\kappa$

1   $x_i^1 = \arg\min_{x_i \in \mathcal{X}_i} h(x_i)$

2   **for** $t = 1, \ldots, T$ **do**

3      Set $A_i^t = (\nabla^2 h(x_i^t) + \eta_t(t+1)\nabla^2 p(x_i^t))^{-1/2}$

4      Play $\hat{x}_i^t = x_i^t + \delta_t A_i^t z_i^t$, receive bandit feedback $c_i(\hat{x}_i, \hat{x}_{-i})$, sample $z_i^t \sim \mathbb{S}^d$

5      Update gradient estimator $\hat{g}_i^t = \frac{d}{\delta_t} c_i(\hat{x}^t)(A_i^t)^{-1} z_i^t$

6      Update the strategy

$$x_i^{t+1} = \arg\min_{x_i \in \mathcal{X}_i} \left\{ \eta_t \langle x_i, \hat{g}_i^t \rangle + \eta_t \kappa(t+1) D_p(x_i, x_i^t) + D_h(x_i, x_i^t) \right\} \quad (1)$$

---

## 5   NO-REGRET CONVERGENCE TO NASH EQUILIBRIUM

In this section, we present our main results on the last-iterate convergence to the Nash equilibrium. We show that Algorithm 1 converges to the Nash equilibrium in monotone, strongly monotone, and linear games. Such convergence is no-regret, meaning that the individual regret of each player is sublinear.

For notational simplicity, we present the results in a perfect bandit feedback model, where player $i$ observes exactly $c_i(x^t)$. The discussion of noisy bandit feedback, where player $i$ observes $c_i(x^t) + \epsilon_i^t$, with $\epsilon_i^t$ be a zero-mean noise, is deferred to the appendix (Theorem D.1).

### 5.1   PERFECT BANDIT FEEDBACK

The following theorem describes the last-iterate convergence rate (in expectation) for convex and strongly convex loss under perfect bandit feedback.

**Theorem 5.1.** *Take* $\eta_t = \begin{cases} \frac{1}{2dt^{3/4}} & \mu = 0 \\ \frac{1}{2dt^{1/2}} & \mu > 0 \end{cases}, \delta_t = \begin{cases} \frac{1}{t^{1/4}} & \mu = 0 \\ 1 & \mu > 0 \end{cases}$. *With Algorithm* 1*, we have*

$$\mathbb{E}\left[ \sum_{i \in \mathcal{N}} D_p\left(x_i^*, x_i^{T+1}\right) \right]$$

$$\leq \begin{cases} O\left( \frac{nd\nu \log(T)}{\kappa T^{1/4}} + \frac{n\zeta dB}{T^{3/4}} + \frac{nBL}{\kappa\sqrt{T}} + \frac{ndC_p}{T^{1/4}} + \frac{nd\log(T)}{\kappa T^{1/4}} + \frac{\sqrt{n}B^2 L \log(T)}{\kappa T^{1/4}} \right) & \mu = 0 \\ O\left( \frac{nd\nu \log(T)}{\kappa\sqrt{T}} + \frac{nd\zeta B}{T} + \frac{nBL}{\kappa\sqrt{T}} + \frac{ndC_p}{\sqrt{T}} + \frac{nd\log(T)}{\kappa\sqrt{T}} + \frac{BL \log(T)}{\mu\kappa\sqrt{T}} \right) & \mu > 0 \end{cases}.$$

In the case of the monotone games, Bravo et al. (2018) showed an asymptotic convergence to Nash equilibrium. To the best of our knowledge, Theorem 5.1 is the first result on the last-iterate convergence rate for monotone games. For strongly monotone games, Bravo et al. (2018) first gave a $O(T^{-1/3})$ last-iterate convergence rate, which was later improved to $O(T^{-1/2})$ by Lin et al. (2021).

While we defer the proof to the appendix, we discuss the main ideas for deriving the results. By the update rule, we can obtain the inequality

$$D_h\left(\omega_i, x_i^{t+1}\right) + \eta_t \kappa(t+1) D_p\left(\omega_i, x_i^{t+1}\right)$$
$$\leq D_h\left(\omega_i, x_i^t\right) + \eta_t \kappa(t+1) D_p\left(\omega_i, x_i^t\right) + \eta_t \left\langle \nabla_i c_i\left(x^t\right), \omega_i - x_i^t \right\rangle + \eta_t \cdot \text{residual terms}, \quad (2)$$

where $\omega_i$ is a fixed point given.

When the game is strongly monotone, we can directly use strongly monotonicity and take $p$ to be the Euclidean norm to obtain a recursive relation similar to $\|\omega_i - x_i^{t+1}\|_2^2 \leq (1 - \eta_t^2)\|\omega_i - x_i^{t+1}\|_2^2 +$ residual terms. This amounts to applying this recursion and upper-binding the residual terms individually to obtain a last-iterate convergence. However, when the game is monotone but not strongly monotone, we will need a different approach. Notice that $G = \nabla c_i - \nabla p$ is a monotone operator. Using the property of Bregman divergence, we have $\langle G(x) - G(x'), x' - x \rangle \leq -\sum_{i \in \mathcal{N}} (D_p(x_i, x_i') + D_p(x_i', x_i))$.

We then sum the recursive inequality and leverage the combination of two regularizations, which obtain

$$\eta_T \kappa (T+1) \sum_{i \in \mathcal{N}} D_p \left( \omega_i, x_i^{T+1} \right)$$

$$\leq \sum_{i \in \mathcal{N}} D_h \left( \omega_i, x_i^1 \right) + \kappa \sum_{i \in \mathcal{N}} D_p \left( \omega_i, x_i^1 \right) + \sum_{t=1}^{T} \sum_{i \in \mathcal{N}} \eta_t \left\langle \nabla_i c_i(\omega), \omega_i - x_i^t \right\rangle$$

$$+ \sum_{t=1}^{T} \sum_{i \in \mathcal{N}} \eta_t \left\langle \hat{g}_i^t - \nabla_i c_i \left( x^t \right), \omega_i - x_i^t \right\rangle + \sum_{t=1}^{T} \sum_{i \in \mathcal{N}} \eta_t \left\langle \hat{g}_i^t, x_i^t - x_i^{t+1} \right\rangle .$$

Now it suffices to properly choose a fixed point $\omega_i$ such that both the first term $\sum_{i \in \mathcal{N}} D_h \left( \omega_i, x_i^1 \right)$ and the third term $\sum_{t=1}^{T} \sum_{i \in \mathcal{N}} \eta_t \left\langle \nabla_i c_i(\omega), \omega_i - x_i^t \right\rangle$ are bounded. When $\omega_i$ is the Nash equilibrium $x_i^*$, the third term can be upper bounded trivially using the monotonicity of $c_i$, while it does not imply a bounded first term. Therefore, we set $\omega_i = x_i^*$ when the first term can be bounded. Otherwise, we set it to a close enough point to $x_i^*$, such that the first term can be bounded and the third term is bounded through a more careful calculation.

**High probability result**    In the case of a strongly monotone game, our results show that the $O(T^{-1/4})$ last-iterate convergence rate holds a high probability. This is the first high-probability result for last-iterate convergence in strongly monotone games.

**Theorem 5.2.** *With a probability of at least* $1 - \log(T)\delta$, $\delta \leq e^{-1}$, *and with Algorithm 1, we have* $\sum_{i \in \mathcal{N}} D_p \left( x_i^*, x_i^{T+1} \right) \leq O \left( \frac{nd\nu \log(T)}{\sqrt{T}} + \frac{nd\zeta B}{T} + \frac{nBL}{\sqrt{T}} + \frac{ndC_p}{\sqrt{T}} + \frac{nd\log(T)}{\sqrt{T}} + \frac{dBL\log(T)}{\mu\sqrt{T}} + \frac{nBd^2 \log^2(1/\delta)\log(T)}{\min\{\sqrt{\mu},\mu\}\sqrt{T}} \right).$

## 5.2 Individual Low Regret

Beyond the fast convergence to Nash equilibrium, our algorithm also ensures each player with a sublinear regret when playing against other players. The sublinear regret convergence is a desirable property as the players could be self-interested in general, and want to ensure their return even when other players are not adhering to the protocol. The low regret property remains true for players that are potentially adversarial, despite the convergence to Nash equilibrium no longer holds in that case.

For player $i$, and a sequence of actions $\{\hat{x}_i^t\}_{t=1}^{T}$, define the individual regret as the cumulative expected difference between the costs received and the cost of playing the hindsight optimal action. That is, $\sum_{t=1}^{T} \mathbb{E} \left[ c_i \left( \hat{x}_i^t, x_{-i}^t \right) - c_i \left( \omega_i, x_{-i}^t \right) \right]$, where $\{x_{-i}^t\}_{t=1}^{T}$ is a fixed sequence of actions of other players. The following theorem shows a guarantee of the individual regret of each player.

**Theorem 5.3.** *Take* $\eta_t = \begin{cases} \frac{1}{2dt^{3/4}} & \mu = 0 \\ \frac{1}{2dt^{1/2}} & \mu > 0 \end{cases}$, $\delta_t = \begin{cases} \frac{1}{t^{1/4}} & \mu = 0 \\ 1 & \mu > 0 \end{cases}$. *For a fixed* $\omega_i \in \mathcal{X}_i$, *a fixed sequence of* $\{x_{-i}^t\}_{t=1}^{T}$, *and with Algorithm 1, we have*

$$\sum_{t=1}^{T} \mathbb{E} \left[ c_i \left( \hat{x}_i^t, x_{-i}^t \right) - c_i \left( \omega_i, x_{-i}^t \right) \right] = \begin{cases} O \left( \nu d T^{3/4} \log(T) + G\sqrt{T} + \ell_i \sqrt{n} B T^{3/4} \right) & \mu = 0 \\ O \left( \nu d \sqrt{T} \log(T) + G\sqrt{T} + \frac{nB\ell_i\sqrt{T}}{\mu} \right) & \mu > 0 \end{cases}.$$

Our result matches the $\sqrt{T}$ regret bound for strongly monotone games (Lin et al., 2021), but applies to monotone games as well.

**Implication on social welfare**    By designing the algorithm to be no-regret, we can also show that the social welfare attained by the algorithm also converges to the optimal value.

The social welfare for a joint action $x$ is defined as $\mathrm{SW}(x) = \sum_{i \in \mathcal{N}} c_i(x)$. We let $\mathrm{OPT} = \min_x \mathrm{SW}(x)$ to denote the optimal social welfare.

**Definition 5.1** (Roughgarden 2015; Syrgkanis et al. 2015). *A game is $(C_1, C_2)$-smooth, $C_1 > 0$, $C_2 < 1$, if there exists a strategy $x'$, such that for any $x \in \mathcal{N}$, $\sum_{i \in \mathcal{N}} c_i(x_i', x_{-i}) \leq C_1 \mathrm{OPT} + C_2 \mathrm{SW}(x)$.*

We have the following proposition which shows that the social welfare converges to optimal welfare on average.

**Proposition 5.1.** *With $\eta_t = \frac{1}{2dt^{3/4}}, \delta_t = \frac{1}{t^{1/4}}$, and suppose every player employ Algorithm 1, we have $\frac{1}{T}\sum_{t=1}^{T} \mathbb{E}\left[\mathrm{SW}(\hat{x})\right] = O\left(\frac{C_1 \mathrm{OPT}}{(1-C_2)} + \frac{n\nu d \log(T)}{(1-C_2)T^{1/4}} + \frac{\sqrt{n}B\sum_{i\in\mathcal{N}}\ell_i}{(1-C_2)T^{1/4}}\right)$.*

### 5.3 SPECIAL CASE: LINEAR COST FUNCTION

When $c_i$ is linear, there does not exist a $p$ that is convex while making $c_i - \kappa p$ convex. Algorithm 1 therefore does not apply to the linear case. This coincides with our intuition that the landscape $c_i$ does not provide enough curvature information for the algorithm to utilize.

To extend the algorithm to the linear case, we modify line 6 of Algorithm 1 as $x_i^{t+1} = \arg\min_{x_i \in \mathcal{X}_i} \{\eta_t \langle x_i, \hat{g}_i^t\rangle + \eta_t \tau(t+1)D_p(x_i, x_i^t) + D_h(x_i, x_i^t)\}$. The idea is to first show the convergence of $x^T$ to a game with the cost $c_i(x) + \tau p(x)$. With this regularized game, we choose $p$ to be a strongly convex function and measure the convergence in terms of the gap function $\langle c_i(x), x_i - x^*\rangle$. By carefully controlling $\tau$, we obtain the following result.

**Theorem 5.4.** *With $\eta_t = \frac{1}{2d\sqrt{t}}$, $\tau = \frac{1}{T^{1/6}}$, $G_p = \sup_x \|\nabla p(x)\|$ and Algorithm 1, we have*

$$\mathbb{E}\left[\sum_{i\in\mathcal{N}} \left\langle \nabla_i c_i\left(x^T\right), x_i^T - x_i^*\right\rangle\right]$$

$$\leq \tilde{O}\left(\frac{BG_p + \sqrt{d(BL+G)(n\nu + nBL + nd^2)}}{T^{1/6}} + \frac{\sqrt{dBL(BL+G)}}{\sqrt{\mu}T^{1/6}} + \frac{\sqrt{dnC_p(BL+G)}}{\sqrt{\mu}T^{1/4}}\right).$$

Similar regularization techniques have been used in the analysis of the zero-sum game (Cen et al., 2021; Cai et al., 2023). Our result matches the last-iterate convergence for zero-sum matrix game (Cai et al., 2023), which is a class of games with linear cost functions. However, our result is more general as it applies to multi-player linear games with convex and compact action sets (while previous works only apply to a simplex action set). It remains open to how games with linear cost functions could be effectively learned and whether the convergence rate could be improved.

## 6 APPLICATION TO TIME-VARYING GAME

In this section, we further apply Algorithm 1 to games that evolve over time. A time-varying game $\mathcal{G}_t$ is a game where the cost function $c_i^t(\cdot)$, $i \in \mathcal{N}$ depends on $t$. The game $\mathcal{G}_t$ is not revealed to the players before choosing their actions $x_t$. We assume that $\mathcal{G}_t$ satisfies Assumption 3.1 for every $t$.

Such evolving games have applications in Kelly's auction and power control, where the cost function may change as time-dependent values change, such as channel gains. While the changes of $\mathcal{G}_t$ can be random, we discuss two cases here, 1) when $\mathcal{G}_t$ converges to a static game $\mathcal{G}$ in $o(T)$ time, and 2) when the variation path of the Nash equilibrium, $\sum_{t=1}^{T}\|x_i^{t+1,*} - x_i^{t,*}\|$ is bounded in $o(T)$.

**Converging monotone game** Let $\mathcal{G}_t$ denote the game formed by the costs $\{c_i^t(\cdot)\}_{i\in\mathcal{N}}$, and $\mathcal{G}$ be the game formed by the costs $\{c_i(\cdot)\}_{i\in\mathcal{N}}$. Suppose $\mathcal{G}_t$ converges to $\mathcal{G}$, and let $x^*$ be the set of Nash equilibrium of the game $\mathcal{G}$. The cost function $c_i^t$ converges to some cost function $c_i$ in $o(T)$ time. The following theorem shows the last iterate convergence to $x^*$.

**Theorem 6.1.** *With $\sum_{t=1}^{T}\sum_{i\in\mathcal{N}}\max_x \|\nabla_i c_i(x) - \nabla_i c_i^t(x)\|_2 = T^\alpha$, take $\eta_t = \frac{1}{2dt^{3/4}}$, $\delta_t = \frac{1}{t^{1/4}}$, and under Algorithm 1, we have $\mathbb{E}\left[\sum_{i\in\mathcal{N}} D_p\left(x_i^*, x_i^{T+1}\right)\right] \leq O\left(\frac{n d\nu \log(T)}{\kappa T^{1/4}} + \frac{n\zeta dB}{T^{3/4}} + \frac{nBL}{\kappa\sqrt{T}} + \frac{ndC_p}{T^{1/4}} + \frac{nd \log(T)}{\kappa T^{1/4}} + \frac{\sqrt{n}B^2 L \log(T)}{\kappa T^{1/4}} + \frac{B}{T^{1/4-\alpha}}\right)$.*

For monotone games, Duvocelle et al. (2023) showed an asymptotic last-iterate convergence rate. To the best of our knowledge, Theorem 6.1 is the first last-iterate convergence rate for the class of converging monotone game.

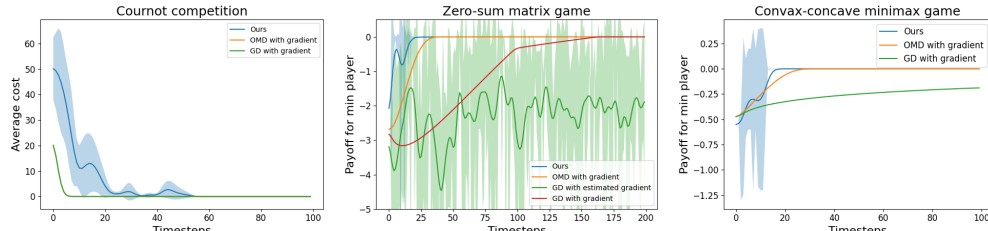

Figure 1: Experiment on Cournot competition, zero-sum two-player minimax game, and convex-concave game. In Cournot competition, the curves of OMD and GD overlap with each other.

**Evolving game and equilibrium tracking**  We now discuss the case where $\mathcal{G}_t$ does not necessarily converge to a game $\mathcal{G}$, but the cumulative changes of the equilibrium are bounded. We use the variation path $V_i(T) = \sum_{t \in [T]} \left\| x_i^{t+1,*} - x_i^{t,*} \right\|$ to track the cumulative changes of equilibrium. In this setting, the last-iterate convergence is not applicable, and the convergence is measured in terms of the average gap. Because of this, the algorithm is slightly modified and updates with $x_i^{t+1} = \arg\min_{x_i \in \mathcal{X}_i} \left\{ \eta_t \langle x_i, \hat{g}_i^t \rangle + D_h(x_i, x_i^t) \right\}$.

**Theorem 6.2.** *Assume* $V_i(T) \leq T^\varphi$, $\varphi \in [0,1]$. *Take* $\eta_t = \frac{1}{2dt^{\frac{(1-\varphi)}{3}}}$, $\delta_t = \frac{1}{t^{1/2}}$, *and under Algorithm 1, we have* $\frac{1}{T} \sum_{t=1}^T \sum_{i \in \mathcal{N}} \left\langle \nabla_i c_i^t \left( \hat{x}_i^t, \hat{x}_{-i}^t \right), \hat{x}_i^t - x_i^{t,*} \right\rangle = \tilde{O}\left( \frac{n\nu d + Ln^{3/2}B^2 + nG}{T^{\frac{2(1-\varphi)}{3}}} + \frac{n}{T^{\frac{9}{8} - \frac{(4\varphi+5)^2}{72}}} \right)$.

In the case of a strongly monotone game, Duvocelle et al. (2023) gave a result of $T^{\varphi/5-1/5}$ and Yan et al. (2023) gave a result of $T^{\varphi/3-2/3}$. In comparison, Theorem 6.2 extends the study to monotone games, and improves the result to $O\left( \max\left\{ T^{2\varphi/3-2/3}, T^{(4\varphi+5)^2/72-9/8} \right\} \right)$.

# 7 EXPERIMENT

In this section, we provide a numerical evaluation of our proposed algorithm in three static games. We repeat each experiment with 5 different random seeds. We ran all experiments with a 10-core CPU, with 32 GB memory. We set $\eta_t = \frac{1}{\sqrt{t+1}}$, and $\delta_t = 0.001$.

We present the results of the following example games described below. More results with other parameters can be found in the Appendix K.

**Cournot competition**  In this Cournot duopoly model, $n$ players compete with constant marginal costs, each having individual constant price intercepts and slopes. We model the game with 5 players, where the margin cost is 40, price intercept is $[30, 50, 30, 50, 30]$, and the price slope is $[50, 30, 50, 30, 50]$.

**Zero-sum matrix game**  In this zero-sum matrix game, the two players aim to solve the bilinear problem $\min_x \max_y x^\top A y$. We set this matrix $A$ to be $[[1, 2], [3, 4]]$.

**monotone zero-sum matrix game**  In this monotone version of the zero-sum matrix game, we regularize the game by the regularizer $x^2 + y^2$.

Algorithm 1 is evaluated against two baseline methods: online mirror descent and gradient descent, with exact gradient, or estimated gradient (bandit feedback). We set the learning rate $\eta$ to be 0.01 in both zero-sum matrix games and monotone zero-sum matrix games and 0.09 in Cournot competition.

Figure 1 summarizes our experimental findings, where our algorithm attains comparable performance to online mirror descent and gradient descent with full information. This demonstrates the efficacy of our algorithm. We also compare our algorithm to gradient descent with an estimated

gradient, using the same ellipsoidal gradient estimator, for a more fair comparison. However, apart from the zero-sum matrix game, we find the baseline algorithm performs too poorly to be compared.

## 8 CONCLUSION

In this work, we present a mirror-descent-based algorithm that converges in $O(T^{-1/4})$ in general monotone and smooth games under bandit feedback and strongly uncoupled dynamics. Our algorithm is no-regret, and the result can be improved to $O(T^{-1/2})$ in the case of strongly-monotone games. To our best knowledge, this is the first uncoupled and convergent algorithm in general monotone games under bandit feedback. We then extend our results to time-varying monotone games and present the first result of $O(T^{-1/4})$ for converging games and the improved result of $O\left(\max\{T^{2\varphi/3-2/3}, T^{(4\varphi+5)^2/72-9/8}\}\right)$ for equilibrium tracking. We further verify the effectiveness of our algorithm with empirical evaluations.

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
