## A  MORE GAME EXAMPLES

**Example A.1** (Extensive form game (EFG)). *EFGs are games on a directed tree. At terminal nodes denoted as $z \in \mathcal{Z}$, each player $i \in \mathcal{N}$ incurs a cost $c_i(z)$ based on a function $c_i : \mathcal{Z} \to \mathbb{R}$. The action set of each player, $\mathcal{X}_i$, is represented through a sequence-form polytope known as $\mathcal{X}_i$ Koller et al. (1996). Considering the probability $p(z)$ of reaching a terminal node $z \in \mathcal{Z}$, the cost for player $i$ is expressed as $c_i(x) := \sum_{z \in \mathcal{Z}} p(z) c_i(z) \prod_{j \in \mathcal{N}} x_j [\sigma_{j,z}]$. Here, $x = (x_1, \ldots, x_n) \in \prod_{j \in \mathcal{N}} \mathcal{X}_j$ signifies the joint strategy profile, and $x_j [\sigma j, z]$ denotes the probability mass assigned to the last sequence $\sigma_{j,z}$ encountered by player $j$ before reaching $z$. The smoothness and concavity of utilities directly arise from multilinearity.*

**Example A.2** (convex potential game). *A game is called a potential game if there exists a potential function $\Phi : \mathcal{X} \to \mathbb{R}$, such that, $c_i(x_i, x_{-i}) - c_i(x_i', x_{-i}) = \Phi(x_i, x_{-i}) - \Phi(x_i', x_{-i})$, for all $i \in \mathcal{N}$. If $\Phi$ is continuous, differentiable, smooth, and convex in $x_i$, then the game satisfies Assumption 3.1. For example, a non-atomic congestion game satisfies Assumption 3.1, as shown in Proposition 1 and 2 of Chen & Lu (2016).*

## B  PROOF OF THEOREM 5.1

**Theorem 5.1.** *Take* $\eta_t = \begin{cases} \frac{1}{2dt^{3/4}} & \mu = 0 \\ \frac{1}{2dt^{1/2}} & \mu > 0 \end{cases}, \delta_t = \begin{cases} \frac{1}{t^{1/4}} & \mu = 0 \\ 1 & \mu > 0 \end{cases}$. *With Algorithm 1, we have*

$$\mathbb{E}\left[\sum_{i \in \mathcal{N}} D_p\left(x_i^*, x_i^{T+1}\right)\right]$$

$$\leq \begin{cases} O\left(\frac{nd\nu \log(T)}{\kappa T^{1/4}} + \frac{n\zeta dB}{T^{3/4}} + \frac{nBL}{\kappa\sqrt{T}} + \frac{ndC_p}{T^{1/4}} + \frac{nd\log(T)}{\kappa T^{1/4}} + \frac{\sqrt{n}B^2L\log(T)}{\kappa T^{1/4}}\right) & \mu = 0 \\ O\left(\frac{nd\nu \log(T)}{\kappa\sqrt{T}} + \frac{nd\zeta B}{T} + \frac{nBL}{\kappa\sqrt{T}} + \frac{ndC_p}{\sqrt{T}} + \frac{nd\log(T)}{\kappa\sqrt{T}} + \frac{BL\log(T)}{\mu\kappa\sqrt{T}}\right) & \mu > 0 \end{cases}.$$

*Proof.* We now upper bound the terms in Lemma J.1.

When $\mu = 0$, taking expectation conditioned on $x^t$, we have $\mathbb{E}\left[\|A_i^t \hat{g}_i^t\|^2 \mid x^t\right] = \frac{d^2}{\delta_t^2}\mathbb{E}\left[c_i(\hat{x}^t)^2 \|z_i^t\|^2 \mid x^t\right] \leq \frac{d^2}{\delta_t^2}$. By Lemma J.2, and the choice $\eta_t = \frac{1}{2d\sqrt{t}}$, we have

$$\sum_{t=1}^{T} \eta_t \sum_{i \in \mathcal{N}} \mathbb{E}\left[\langle \hat{g}_i^t, x_i^t - x_i^{t+1}\rangle\right] \leq \sum_{t=1}^{T} \eta_t^2 \sum_{i \in \mathcal{N}} \mathbb{E}\left[\|A_i^t \hat{g}_i^t\|^2\right] \leq nd^2 \sum_{t=1}^{T} \frac{\eta_t^2}{\delta_t^2}.$$

By the definition of $\hat{c}_i$,

$$\sum_{i \in \mathcal{N}} \sum_{t=1}^{T} \eta_t \mathbb{E}\left[\langle \hat{g}_i^t - \nabla_i c_i\left(x^t\right), \omega_i - x_i^t\rangle \mid x^t\right]$$

$$= \sum_{i \in \mathcal{N}} \sum_{t=1}^{T} \eta_t \mathbb{E}\left[\langle \nabla_i \hat{c}_i(x^t) - \nabla_i c_i\left(x^t\right), \omega_i - x_i^t\rangle \mid x^t\right]$$

$$= \sum_{i \in \mathcal{N}} \sum_{t=1}^{T} \eta_t \mathbb{E}\left[\mathbb{E}_{w_i \sim \mathbb{B}^d}\mathbb{E}_{\mathbf{z}_{-i} \sim \Pi_{j \neq i}\mathbb{S}^d} \left\langle \nabla_i c_i\left(x_i^t + \delta_t A_i^t w_i, \hat{x}_{-i}^t\right) - \nabla_i c_i\left(x^t\right), \omega_i - x_i^t\right\rangle \mid x^t\right]$$

$$\leq B \sum_{i \in \mathcal{N}} \sum_{t=1}^{T} \eta_t \mathbb{E}\left[\mathbb{E}_{w_i \sim \mathbb{B}^d}\mathbb{E}_{\mathbf{z}_{-i} \sim \Pi_{j \neq i}\mathbb{S}^d} \left\|\nabla_i c_i\left(x_i^t + \delta_t A_i^t w_i, \hat{x}_{-i}^t\right) - \nabla_i c_i\left(x^t\right)\right\| \mid x^t\right]$$

By the smoothness of $c_i$,

$$\mathbb{E}_{w_i \sim \mathbb{B}^d}\mathbb{E}_{\mathbf{z}_{-i} \sim \Pi_{j \neq i}\mathbb{S}^d}\left[\left\|\nabla_i c_i\left(x_i^t + \delta_t A_i^t w_i, \hat{x}_{-i}^t\right) - \nabla_i c_i\left(x^t\right)\right\|\right]$$

$$\leq \ell_i \mathbb{E}_{w_i \sim \mathbb{B}^d}\mathbb{E}_{\mathbf{z}_{-i} \sim \Pi_{j \neq i}\mathbb{S}^d}\left[\sqrt{\delta_t^2 \|A_i w_i\|^2 + \delta_t^2 \sum_{j \neq i} \|A_j z_j\|^2}\right].$$

Since $p$ is convex, $\nabla^2 p(x)$ is positive semi-definite, and $A_i^t \preceq (\nabla^2 h(x_i))^{-1/2}$. For $\bar{x}_i^t = x_i^t + A_i^t w_i^t$. Define $\|v\|_x = \sqrt{v^\top \nabla^2 h(x)v}$, we have $\|\bar{x}_i^t - x_i^t\|_{x_i} \leq \|\omega_i^t\| \leq 1$, and $\bar{x}_i^t \in W(x_i^t)$, where $W(x_i) = \{x_i' \in \mathbb{R}^d, \|x_i' - x_i\|_{x_i} \leq 1\}$ is the Dikin ellipsoid. Since $W(x_i) \subseteq \mathcal{X}_i, \forall x_i \in int(\mathcal{X}_i)$, we can upper bound $\|A_i w_i\|^2$ by $B^2$, the diameter of the set $\mathcal{X}_i$. Hence $\|\nabla_i \hat{c}_i(x^t) - \nabla_i c_i\left(x^t\right)\| \leq \ell_i \delta_t \sqrt{n}B$. By Lemma J.5

$$\sum_{i \in \mathcal{N}} \sum_{t=1}^{T} \eta_t \mathbb{E}\left[\langle \hat{g}_i^t - \nabla_i c_i\left(x^t\right), \omega_i - x_i^t\rangle \mid x^t\right] = \sum_{i \in \mathcal{N}} \sum_{t=1}^{T} \eta_t \mathbb{E}\left[\langle \nabla_i \hat{c}_i\left(x^t\right) - \nabla_i c_i\left(x^t\right), \omega_i - x_i^t\rangle \mid x^t\right]$$

$$\leq \sum_{i \in \mathcal{N}} \sum_{t=1}^{T} \eta_t \mathbb{E}\left[\left\|\nabla_i \hat{c}_i\left(x^t\right) - \nabla_i c_i(x^t)\right\| \left\|\omega_i - x_i^t\right\| \mid x^t\right]$$

$$\leq \sqrt{n}B^2 \sum_{i \in \mathcal{N}} \ell_i \sum_{t=1}^{T} \eta_t \delta_t.$$

When $\mu > 0$, we set $\delta = 1$. Then, taking expectation conditioned on $x^t$, we have $\mathbb{E}\left[\|A_i^t \hat{g}_i^t\|^2 \mid x^t\right] = d^2 \mathbb{E}\left[c_i(\hat{x}^t)^2 \|z_i^t\|^2 \mid x^t\right] \leq d^2$. By Lemma J.2, and the choice $\eta_t = \frac{1}{2d\sqrt{t}}$, we have

$$\sum_{t=1}^{T} \eta_t \sum_{i \in \mathcal{N}} \mathbb{E}\left[\langle \hat{g}_i^t, x_i^t - x_i^{t+1} \rangle\right] \leq \sum_{t=1}^{T} \eta_t^2 \sum_{i \in \mathcal{N}} \mathbb{E}\left[\|A_i^t \hat{g}_i^t\|^2\right] \leq n d^2 \sum_{t=1}^{T} \eta_t^2.$$

By Lemma J.5, for any $\omega_i \in \mathcal{X}_i$, we have

$$\sum_{i \in \mathcal{N}} \sum_{t=1}^{T} \eta_t \mathbb{E}\left[\langle \hat{g}_i^t - \nabla_i c_i\left(x^t\right), \omega_i - x_i^t \rangle \mid x^t\right] = \sum_{i \in \mathcal{N}} \sum_{t=1}^{T} \eta_t \mathbb{E}\left[\langle \nabla_i \hat{c}_i(x^t) - \nabla_i c_i\left(x^t\right), \omega_i - x_i^t \rangle \mid x^t\right]$$

$$\leq \sum_{i \in \mathcal{N}} B \ell_i \sum_{t=1}^{T} \eta_t \mathbb{E}\left[\sum_{j \in \mathcal{N}} \left(\sigma_{\max}\left(A_j^t\right)^2\right) \mid x^t\right]$$

$$\leq \sum_{i \in \mathcal{N}} B \ell_i \sum_{t=1}^{T} \frac{1}{\mu(t+1)}$$

$$\leq \frac{B \sum_{i \in \mathcal{N}} \ell_i}{\mu} \sum_{t=1}^{T} \frac{1}{(t+1)}.$$

where the third inequality is by $\nabla^2 h(x)$ being positive definite, and $\nabla^2 p(x) \geq \mu I$.

Let $L = \sum_{i \in \mathcal{N}} \ell_i$. When $\mu = 0$, combing and rearranging the terms, we have

$$\mathbb{E}\left[\sum_{i \in \mathcal{N}} D_p\left(x_i^*, x_i^{T+1}\right)\right]$$

$$\leq O\left(\frac{n\nu \log(T)}{\kappa \eta_T T} + \frac{n\zeta B}{\eta_T T^{3/2}} + \frac{nBL}{\kappa \sqrt{T}} + \frac{n}{\kappa \sqrt{T}} + \frac{nC_p}{\eta_T T} + \frac{nd^2}{\kappa \eta_T T} \sum_{t=1}^{T} \frac{\eta_t^2}{\delta_t^2} + \frac{\sqrt{n} B^2 L \sum_{t=1}^{T} \eta_t \delta_t}{\kappa \eta_T T}\right).$$

Take $\eta_t = \frac{1}{2d t^{3/4}}$, $\delta_t = \frac{1}{t^{1/4}}$, then $\sum_{t=1}^{T} \frac{\eta_t^2}{\delta_t^2} = O\left(\sum_{t=1}^{T} \frac{1}{t}\right) = O(\log(T))$, and $\sum_{t=1}^{T} \eta_t \delta_t = O\left(\sum_{t=1}^{T} \frac{1}{t}\right) = O(\log(T))$. Hence, we have

$$\mathbb{E}\left[\sum_{i \in \mathcal{N}} D_p\left(x_i^*, x_i^{T+1}\right)\right] \leq O\left(\frac{nd\nu \log(T)}{\kappa T^{1/4}} + \frac{n\zeta dB}{T^{3/4}} + \frac{nBL}{\kappa \sqrt{T}} + \frac{ndC_p}{T^{1/4}} + \frac{nd \log(T)}{\kappa T^{1/4}} + \frac{\sqrt{n} B^2 L \log(T)}{\kappa T^{1/4}}\right).$$

When $\mu > 0$, combing and rearranging the terms, we have

$$\mathbb{E}\left[\sum_{i \in \mathcal{N}} D_p\left(x_i^*, x_i^{T+1}\right)\right]$$

$$\leq O\left(\frac{n\nu \log(T)}{\kappa \eta_T T} + \frac{n\zeta B}{\eta_T T^{3/2}} + \frac{nBL}{\kappa \sqrt{T}} + \frac{n}{\sqrt{T}} + \frac{nC_p}{\eta_T T} + \frac{nd^2}{\kappa \eta_T T} \sum_{t=1}^{T} \eta_t^2 + \frac{BL \log(T)}{\mu \kappa \eta_T T}\right).$$

Take $\eta_t = \frac{1}{2d t^{1/2}}$, we have

$$\mathbb{E}\left[\sum_{i \in \mathcal{N}} D_p\left(x_i^*, x_i^{T+1}\right)\right] \leq O\left(\frac{nd\nu \log(T)}{\kappa \sqrt{T}} + \frac{nd\zeta B}{T} + \frac{nBL}{\kappa \sqrt{T}} + \frac{ndC_p}{\sqrt{T}} + \frac{nd \log(T)}{\kappa \sqrt{T}} + \frac{BL \log(T)}{\mu \kappa \sqrt{T}}\right).$$

$\square$

## C   PROOF OF THEOREM 5.3

**Theorem 5.3.** *Take* $\eta_t = \begin{cases} \frac{1}{2dt^{3/4}} & \mu = 0 \\ \frac{1}{2dt^{1/2}} & \mu > 0 \end{cases}$, $\delta_t = \begin{cases} \frac{1}{t^{1/4}} & \mu = 0 \\ 1 & \mu > 0 \end{cases}$. *For a fixed* $\omega_i \in \mathcal{X}_i$, *a fixed sequence of* $\{x^t_{-i}\}_{t=1}^T$, *and with Algorithm 1, we have*

$$
\sum_{t=1}^{T} \mathbb{E}\left[c_i\left(\hat{x}^t_i, x^t_{-i}\right) - c_i\left(\omega_i, x^t_{-i}\right)\right] = \begin{cases} O\left(\nu d T^{3/4}\log(T) + G\sqrt{T} + \ell_i\sqrt{n}BT^{3/4}\right) & \mu = 0 \\ O\left(\nu d\sqrt{T}\log(T) + G\sqrt{T} + \frac{nB\ell_i\sqrt{T}}{\mu}\right) & \mu > 0 \end{cases}.
$$

*Proof.* Define the smoothed version of $c_i$ as $\hat{c}_i(x) = \mathbb{E}_{w_i \sim \mathbb{B}^d}\left[c_i\left(x_i + \delta A_i w_i, x_{-i}\right)\right]$. Then, we decompose as

$$
\sum_{t=1}^{T} c_i\left(\hat{x}^t_i, x^t_{-i}\right) - c_i\left(\omega_i, x^t_{-i}\right) = \sum_{t=1}^{T}\left(\hat{c}_i\left(x^t_i, x^t_{-i}\right) - \hat{c}_i\left(\omega_i, x^t_{-i}\right)\right) + \sum_{t=1}^{T}\left(c_i\left(x^t_i, x^t_{-i}\right) - \hat{c}_i\left(x^t_i, x^t_{-i}\right)\right)
$$

$$
+ \sum_{t=1}^{T}\left(\hat{c}_i\left(\omega_i, x^t_{-i}\right) - c_i\left(\omega_i, x^t_{-i}\right)\right) + \sum_{t=1}^{T}\left(c_i\left(\hat{x}^t_i, x^t_{-i}\right) - c_i\left(x^t_i, x^t_{-i}\right)\right).
$$

For the first term, recall that by the update rule, we have,

$$
D_h\left(\omega_i, x^{t+1}_i\right) + \eta_t\kappa(t+1)D_p\left(\omega_i, x^{t+1}_i\right)
$$
$$
= D_h\left(\omega_i, x^t_i\right) + \eta_t\kappa(t+1)D_p\left(\omega_i, x^t_i\right) + \eta_t\left\langle\nabla\hat{c}_i\left(x^t\right), \omega_i - x^t_i\right\rangle + \eta_t\left\langle\hat{g}^t_i - \nabla\hat{c}_i\left(x^t\right), \omega_i - x^t_i\right\rangle
$$
$$
+ \eta_t\left\langle\hat{g}^t_i, x^t_i - x^{t+1}_i\right\rangle
$$
$$
= D_h\left(\omega_i, x^t_i\right) + \eta_t\kappa(t+1)D_p\left(\omega_i, x^t_i\right) + \eta_t\left\langle\nabla\hat{c}_i\left(x^t\right) - \kappa\nabla p(x^t_i), \omega_i - x^t_i\right\rangle + \eta_t\left\langle\hat{g}^t_i - \nabla\hat{c}_i\left(x^t\right) + \kappa\nabla p(x^t_i), \omega_i - x^t_i\right\rangle
$$
$$
+ \eta_t\left\langle\hat{g}^t_i, x^t_i - x^{t+1}_i\right\rangle.
$$

By Lemma J.5, for any $\omega_i \in \mathcal{X}_i$, we have

$$
\mathbb{E}\left[\left\langle\hat{g}^t_i - \nabla\hat{c}_i\left(x^t\right) + \kappa\nabla p(x^t_i), \omega_i - x^t_i\right\rangle \mid x^t\right] = \mathbb{E}\left[\left\langle\nabla_i\hat{c}_i(x^t) - \nabla_i\hat{c}_i\left(x^t\right) + \kappa\nabla p(x^t_i), \omega_i - x^t_i\right\rangle \mid x^t\right]
$$
$$
= \mathbb{E}\left[\kappa\left\langle\nabla p(x^t_i), \omega_i - x^t_i\right\rangle \mid x^t\right]
$$
$$
= \mathbb{E}\left[\kappa p(\omega_i) - \kappa p(x^t_i) - \kappa D_p(\omega_i, x^t_i) \mid x^t\right],
$$

where the last equality follows from the definition of Bregman divergence.

Therefore,

$$
\mathbb{E}\left[D_h\left(\omega_i, x^{t+1}_i\right) + \eta_t\kappa(t+1)D_p\left(\omega_i, x^{t+1}_i\right)\right]
$$
$$
= \mathbb{E}\left[D_h\left(\omega_i, x^t_i\right) + \eta_t\kappa t D_p\left(\omega_i, x^t_i\right) + \eta_t\left\langle\nabla\hat{c}_i\left(x^t\right) - \kappa\nabla p(x^t_i), \omega_i - x^t_i\right\rangle\right] + \eta_t\mathbb{E}\left[\kappa p(\omega_i) - \kappa p(x^t_i)\right]
$$
$$
+ \mathbb{E}\left[\eta_t\left\langle\hat{g}^t_i, x^t_i - x^{t+1}_i\right\rangle\right].
$$

By the monotonicity of $\hat{c}_i\left(x^t\right) - \kappa p(x^t_i)$, we have

$$
\left\langle\nabla\hat{c}_i\left(x^t\right) - \kappa\nabla p(x^t_i), \omega_i - x^t_i\right\rangle \le \left(\hat{c}_i\left(\omega_i, x^t_{-i}\right) - \kappa p(\omega_i)\right) - \left(\hat{c}_i\left(x^t_i, x^t_{-i}\right) - \kappa p(x^t_i)\right).
$$

Hence

$$
\mathbb{E}\left[\hat{c}_i\left(x^t_i, x^t_{-i}\right) - \hat{c}_i\left(\omega_i, x^t_{-i}\right)\right]
$$
$$
\le \mathbb{E}\left[\frac{\left(D_h\left(\omega_i, x^t_i\right) - D_h\left(\omega_i, x^{t+1}_i\right)\right)}{\eta_t} + \kappa\left(tD_p\left(\omega_i, x^t_i\right) - (t+1)D_p\left(\omega_i, x^{t+1}_i\right)\right) + \left\langle\hat{g}^t_i, x^t_i - x^{t+1}_i\right\rangle\right].
$$

When $\mu = 0$, by Lemma J.2, we have $\mathbb{E}\left[\left\langle\hat{g}^t_i, x^t_i - x^{t+1}_i\right\rangle\right] \le \eta_t\mathbb{E}\left[\|A^t_i\hat{g}^t_i\|^2\right]$. Taking expectation conditioned on $x^t$, we have $\mathbb{E}\left[\|A^t_i\hat{g}^t_i\|^2 \mid x^t\right] = \frac{d^2}{\delta^2_t}\mathbb{E}\left[\tilde{c}_i(\hat{x}^t)^2\|z^t_i\|^2 \mid x^t\right] \le \frac{d^2}{\delta^2_t}$, and therefore $\mathbb{E}\left[\left\langle\hat{g}^t_i, x^t_i - x^{t+1}_i\right\rangle\right] \le \frac{\eta_t d^2}{\delta^2_t}$.

Taking summation over $T$, and take $\eta_t = \frac{1}{2dt^{3/4}}$, $\delta_t = \frac{1}{t^{1/4}}$ we have

$$\sum_{t=1}^{T} \mathbb{E}\left[\hat{c}_i\left(x_i^t, x_{-i}^t\right) - \hat{c}_i\left(\omega_i, x_{-i}^t\right)\right] \leq dT^{3/4}\mathbb{E}\left[D_h\left(\omega_i, x_i^1\right)\right] + \kappa\mathbb{E}\left[D_p\left(\omega_i, x_i^1\right)\right] + \sum_{t=1}^{T}\frac{\eta_t d^2}{\delta^2}$$

$$\leq O\left(dT^{3/4}\mathbb{E}\left[D_h\left(\omega_i, x_i^1\right)\right] + \kappa C_p + T^{3/4}\right),$$

as we assumed $D_p(x_i, x_i')$ is bounded for any $x_i, x_i'$.

When $\mu > 0$, taking expectation conditioned on $x^t$, we have $\mathbb{E}\left[\|A_i^t \hat{g}_i^t\|^2 \mid x^t\right] = d^2\mathbb{E}\left[c_i(\hat{x}^t)^2\|z_i^t\|^2 \mid x^t\right] \leq d^2$. By Lemma J.2, and the choice $\eta_t = \frac{1}{2d\sqrt{t}}$, we have

$$\sum_{t=1}^{T}\sum_{i\in\mathcal{N}}\mathbb{E}\left[\langle\hat{g}_i^t, x_i^t - x_i^{t+1}\rangle\right] \leq \sum_{t=1}^{T}\eta_t\sum_{i\in\mathcal{N}}\mathbb{E}\left[\|A_i^t\hat{g}_i^t\|^2\right] \leq nd^2\sum_{t=1}^{T}\eta_t = nd^2\sqrt{T}.$$

Taking summation over $T$, and take $\eta_t = \frac{1}{2dt^{1/2}}$, we have

$$\sum_{t=1}^{T}\mathbb{E}\left[\hat{c}_i\left(x_i^t, x_{-i}^t\right) - \hat{c}_i\left(\omega_i, x_{-i}^t\right)\right] \leq dT^{1/2}\mathbb{E}\left[D_h\left(\omega_i, x_i^1\right)\right] + \kappa\mathbb{E}\left[D_p\left(\omega_i, x_i^1\right)\right] + nd^2\sqrt{T},$$

as we assumed $D_p(x_i, x_i')$ is bounded for any $x_i, x_i'$.

Define $\pi_x(y) = \inf\left\{t \geq 0 : x + \frac{1}{t}(y - x) \in \mathcal{X}_i\right\}$. Notice that $x_i^1(x) = \arg\min_{x_i\in\mathcal{X}_i} h(x_i)$, so $D_h(\omega_i, x_i^1) = h(\omega_i) - h(x_i^1)$.

- If $\pi_{x_i^1}(\omega_i) \leq 1 - \frac{1}{\sqrt{T}}$, then by Lemma J.6, $D_h(\omega_i, x_i^1) = \nu\log(T)$, and $\sum_{t=1}^{T}\mathbb{E}\left[\hat{c}_i\left(x_i^t, x_{-i}^t\right) - \hat{c}_i\left(\omega_i, x_{-i}^t\right)\right] = O\left(\nu dT^{3/4}\log(T)\right)$.

- Otherwise, we find a point $\omega_i'$ such that $\|\omega_i' - \omega_i\| = O(1/\sqrt{T})$ and $\pi_{x_i^1}(\omega_i') \leq 1 - \frac{1}{\sqrt{T}}$. Then $D_h(\omega_i', x_i^1) = \nu\log(T)$,

$$\hat{c}_i\left(\omega_i', x_{-i}^t\right) - \hat{c}_i\left(\omega_i, x_{-i}^t\right) \leq \langle\nabla_i\hat{c}_i\left(\omega_i', x_{-i}^t\right), \omega_i' - \omega_i\rangle \leq \|\nabla_i\hat{c}_i\left(\omega_i', x_{-i}^t\right)\|\|\omega_i' - \omega_i\| \leq \frac{\max_x\|\nabla_i c_i\left(x\right)\|}{\sqrt{T}}.$$

Therefore, $\sum_{t=1}^{T}\mathbb{E}\left[\hat{c}_i\left(x_i^t, x_{-i}^t\right) - \hat{c}_i\left(\omega_i, x_{-i}^t\right)\right] = O\left(\nu dT^{3/4}\log(T) + \max_x\|\nabla_i c_i\left(x\right)\|\sqrt{T}\right)$.

For the second term, by Jensen's inequality, we have

$$\hat{c}_i\left(x_i^t, x_{-i}^t\right)\mathbb{E}_{w_i^t\sim\mathbb{B}^d}\left[c_i\left(x_i^t + \delta_t A_i^t w_i^t, x_{-i}^t\right)\right] \geq c_i\left(\mathbb{E}_{w_i^t\sim\mathbb{B}^d}x_i^t + \delta_t A_i^t w_i^t, x_{-i}^t\right) = c_i\left(x_i^t, x_{-i}^t\right).$$

Therefore, we have $\sum_{t=1}^{T}\left(c_i\left(x_i^t, x_{-i}^t\right) - \hat{c}_i\left(x_i^t, x_{-i}^t\right)\right) = 0$.

When $\mu = 0$, by the definition of $\hat{c}_i$ and the smoothness of $c_i$,

$$\|\nabla_i\hat{c}_i(x^t) - \nabla_i c_i\left(x^t\right)\| = \left\|\mathbb{E}_{w_i\sim\mathbb{B}^d}\mathbb{E}_{\mathbf{z}_{-i}\sim\Pi_{j\neq i}\mathbb{S}^d}\left[\nabla_i c_i\left(x_i^t + \delta_t A_i^t w_i, \hat{x}_{-i}^t\right) - \nabla_i c_i\left(x^t\right)\right]\right\|$$

$$\leq \ell_i\sqrt{\mathbb{E}_{w_i\sim\mathbb{B}^d}\mathbb{E}_{\mathbf{z}_{-i}\sim\Pi_{j\neq i}\mathbb{S}^d}\left[\delta_t^2\|\delta_t A_i w_i\|^2 + \delta_t^2\sum_{j\neq i}\|A_j z_j\|^2\right]}.$$

Since $p$ is convex, $\nabla^2 p(x)$ is positive semi-definite, and $A_i^t \preceq (\nabla^2 h(x_i))^{-1/2}$. For $\bar{x}_i^t = x_i^t + A_i^t w_i^t$. Define $\|v\|_x = \sqrt{v^\top\nabla^2 h(x)v}$, we have $\|\bar{x}_i^t - x_i^t\|_{x_i} \leq \|\omega_i^t\| \leq 1$, and $\bar{x}_i^t \in W(x_i^t)$, where $W(x) = \{x_i' \in \mathbb{R}^d, \|x_i' - x_i\|_{x_i} \leq 1\}$ is the Dikin ellipsoid. Since $W(x_i) \subseteq \mathcal{X}_i, \forall x_i \in int(\mathcal{X}_i)$, we can upper bound $\|A_i w_i\|^2$ by $B^2$, the diameter of the set $\mathcal{X}_i$. Hence $\|\nabla_i\hat{c}_i(x^t) - \nabla_i c_i\left(x^t\right)\| \leq \ell_i\delta_t\sqrt{n}B$.

Therefore, for the third term, we have

$$\sum_{t=1}^{T} \mathbb{E}\left[\hat{c}_i\left(\omega_i, x_{-i}^t\right) - c_i\left(\omega_i, x_{-i}^t\right)\right] \le O\left(\sum_{t=1}^{T} \ell_i \delta_t \sqrt{n} B\right).$$

Similarly, for the fourth term, we have $\sum_{t=1}^{T} \mathbb{E}\left[c_i\left(\hat{x}_i^t, x_{-i}^t\right) - c_i\left(x_i^t, x_{-i}^t\right)\right] \le O\left(\sum_{t=1}^{T} \ell_i \delta_t \sqrt{n} B\right)$.

When $\mu > 0$, by Lemma J.5, for any $\omega_i \in \mathcal{X}_i$, we have

$$\left\|\nabla_i \hat{c}_i(x^t) - \nabla_i c_i\left(x^t\right)\right\| \le \ell_i \sqrt{\sum_{j \in \mathcal{N}} \left(\sigma_{\max}\left(A_j^t\right)^2\right)} \le \frac{n\ell_i}{\sqrt{\mu(t+1)}}.$$

where the second inequality is by $\nabla^2 h(x)$ being positive definite, and $\nabla^2 p(x) \ge \mu I$.

Therefore, for the third term, we have

$$\sum_{t=1}^{T} \mathbb{E}\left[\hat{c}_i\left(\omega_i, x_{-i}^t\right) - c_i\left(\omega_i, x_{-i}^t\right)\right] \le O\left(\frac{nB\ell_i \sqrt{T}}{\mu}\right).$$

Similarly, for the fourth term, we have $\sum_{t=1}^{T} \mathbb{E}\left[c_i\left(\hat{x}_i^t, x_{-i}^t\right) - c_i\left(x_i^t, x_{-i}^t\right)\right] \le O\left(\frac{nB\ell_i \sqrt{T}}{\mu}\right)$.

When $\mu = 0$, with $\delta_t = \frac{1}{t^{1/4}}$, we have the regret as

$$\sum_{t=1}^{T} \mathbb{E}\left[c_i\left(\hat{x}_i^t, x_{-i}^t\right) - c_i\left(\omega_i, x_{-i}^t\right)\right] = O\left(\nu d T^{3/4} \log(T) + \max_x \|\nabla_i c_i\left(x\right)\| \sqrt{T} + \ell_i \sqrt{n} B T^{3/4}\right).$$

When $\mu > 0$, we have the regret as

$$\sum_{t=1}^{T} \mathbb{E}\left[c_i\left(\hat{x}_i^t, x_{-i}^t\right) - c_i\left(\omega_i, x_{-i}^t\right)\right] = O\left(\nu d T^{1/2} \log(T) + \max_x \|\nabla_i c_i\left(x\right)\| \sqrt{T} + \frac{nB\ell_i \sqrt{T}}{\mu}\right).$$

Combining the terms yields the final result. $\qquad\square$

# D PROOF OF THEOREM D.1

We now consider the case where every player receive $\tilde{c}_i(x^t) = c_i(x^t) + \epsilon_i^t$, where $\mathbb{E}[\epsilon_i^t \mid \hat{x}^t] = 0$, and $\|\epsilon_i^t\|^2 \leq \sigma$. The following theorem describes the last-iterate convergence rate (in expectation) for monotone and strongly monotone games under noisy bandit feedback.

**Theorem D.1.** *With* $\eta_t = \frac{1}{4d^2(1+\sigma)t^{3/4}}$, $\delta_t = \frac{1}{t^{1/4}}$

$$\sum_{i \in \mathcal{N}} D_p\left(x_i^*, x_i^{T+1}\right) \leq O\left(\frac{n\nu d^2(1+\sigma)\log(T)}{\kappa T^{1/4}} + \frac{n\zeta d^2(1+\sigma)B}{T^{3/4}} + \frac{nd^2(1+\sigma)C_p}{T^{1/4}}\right.$$

$$\left. + \frac{\sqrt{n}B^2 L\log(T)}{\kappa T^{1/4}} + \frac{nd\log(T)}{\kappa(1+\sigma)^2 T^{1/4}}\right).$$

*Proof.* Similar to Theorem 5.1, with Lemma J.1, we have

$$\sum_{i \in \mathcal{N}} D_p\left(x_i^*, x_i^{T+1}\right) \leq O\left(\frac{n\nu\log(T)}{\kappa\eta_T T} + \frac{n\zeta B}{\eta_T T^{3/2}}\right) + O\left(\frac{nB\sum_{i \in \mathcal{N}}\ell_i}{\kappa T^{3/2}} + \frac{n}{\kappa T^{3/2}}\right)\frac{\sum_{t=1}^{T}\eta_t}{\eta_T} + O\left(\frac{nC_p}{\eta_T T}\right)$$

$$+ \frac{\sqrt{n}B^2 L\sum_{t=1}^{T}\eta_t\delta_t}{\eta_T\kappa(T+1)} + \frac{1}{\eta_T\kappa(T+1)}\sum_{i \in \mathcal{N}}\sum_{t=1}^{T}\eta_t\left\langle \hat{g}_i^t, x_i^t - x_i^{t+1}\right\rangle.$$

Taking expectation conditioned on $x^t$, we have $\mathbb{E}\left[\|A_i^t\hat{g}_i^t\|^2 \mid x^t\right] = \frac{d^2}{\delta_t^2}\mathbb{E}\left[\tilde{c}_i(\hat{x}^t)^2\|z_i^t\|^2 \mid x^t\right] \leq \frac{d^2}{\delta_t^2}(2+2\sigma)$. By Lemma J.2, and the choice $\eta_t = \frac{1}{4d^2(1+\sigma)t^{3/4}}$, we have

$$\sum_{t=1}^{T}\eta_t\sum_{i \in \mathcal{N}}\mathbb{E}\left[\left\langle \hat{g}_i^t, x_i^t - x_i^{t+1}\right\rangle\right] \leq \sum_{t=1}^{T}\eta_t^2\sum_{i \in \mathcal{N}}\mathbb{E}\left[\|A_i^t\hat{g}_i^t\|^2\right] \leq nd^2\sum_{t=1}^{T}\frac{\eta_t^2}{\delta_t^2} = \frac{n\log(T)}{16(1+\sigma)^2}.$$

Combining everything, we have

$$\sum_{i \in \mathcal{N}} D_p\left(x_i^*, x_i^{T+1}\right)$$

$$\leq O\left(\frac{n\nu d^2(1+\sigma)\log(T)}{\kappa T^{1/4}} + \frac{n\zeta d^2(1+\sigma)B}{T^{3/4}} + \frac{nd^2(1+\sigma)C_p}{T^{1/4}} + \frac{\sqrt{n}B^2 L\log(T)}{\kappa T^{1/4}} + \frac{nd\log(T)}{\kappa(1+\sigma)^2 T^{1/4}}\right).$$

$$\square$$

# E PROOF OF THEOREM 5.2

**Theorem 5.2.** *With a probability of at least $1 - \log(T)\delta$, $\delta \leq e^{-1}$, and with Algorithm 1, we have $\sum_{i \in \mathcal{N}} D_p\left(x_i^*, x_i^{T+1}\right) \leq O\left(\frac{nd\nu \log(T)}{\sqrt{T}} + \frac{nd\zeta B}{T} + \frac{nBL}{\sqrt{T}} + \frac{ndC_p}{\sqrt{T}} + \frac{nd \log(T)}{\sqrt{T}} + \frac{dBL \log(T)}{\mu\sqrt{T}} + \frac{nBd^2 \log^2(1/\delta) \log(T)}{\min\{\sqrt{\mu}, \mu\}\sqrt{T}}\right).$*

*Proof.* Lemma J.1, we have

$$\sum_{i \in \mathcal{N}} D_p\left(x_i^*, x_i^{T+1}\right)$$

$$\leq O\left(\frac{n\nu \log(T)}{\kappa \eta_T T} + \frac{n\zeta B}{\eta_T T^{3/2}}\right) + O\left(\frac{nB \sum_{i \in \mathcal{N}} \ell_i}{\kappa T^{3/2}} + \frac{n}{\kappa T^{3/2}}\right) \frac{\sum_{t=1}^T \eta_t}{\eta_T} + O\left(\frac{nC_p}{\eta_T T}\right)$$

$$+ \frac{1}{\kappa \eta_T (T+1)} \sum_{i \in \mathcal{N}} \sum_{t=1}^T \eta_t \left\langle \hat{g}_i^t, x_i^t - x_i^{t+1} \right\rangle + \frac{1}{\kappa \eta_T (T+1)} \sum_{t=1}^T \eta_t \sum_{i \in \mathcal{N}} \left\langle \hat{g}_i^t - \nabla_i c_i\left(x^t\right), \omega_i - x_i^t \right\rangle.$$

By Lemma J.2, we have

$$\sum_{t=1}^T \eta_t \sum_{i \in \mathcal{N}} \left\langle \hat{g}_i^t, x_i^t - x_i^{t+1} \right\rangle \leq \sum_{t=1}^T \eta_t^2 \sum_{i \in \mathcal{N}} \left\|A_i^t \hat{g}_i^t\right\|^2 \leq nd^2 \sum_{t=1}^T \eta_t^2.$$

We then decompose the last term as

$$\sum_{t=1}^T \eta_t \sum_{i \in \mathcal{N}} \left\langle \hat{g}_i^t - \nabla_i c_i\left(x^t\right), \omega_i - x_i^t \right\rangle = \sum_{t=1}^T \eta_t \sum_{i \in \mathcal{N}} \left\langle g_i^t - \hat{c}_i^t(x_i^t), \omega_i - x_i^t \right\rangle + \sum_{i \in \mathcal{N}} \sum_{t=1}^T \eta_t \left\langle \nabla_i \hat{c}_i(x^t) - \nabla_i c_i\left(x^t\right), \omega_i - x_i^t \right\rangle.$$

By Lemma E.1, we have

$$\sum_{t=1}^T \eta_t \left\langle g_i^t - \hat{c}_i^t(x_i^t), \omega_i - x_i^t \right\rangle \leq O\left(\frac{Bd \log^2(1/\delta) \log(T)}{\min\{\sqrt{\mu}, \mu\}}\right),$$

with a probability of at least $1 - \log(T)\delta$, $\delta \leq e^{-1}$.

By Lemma J.5, for any $\omega_i \in \mathcal{X}_i$, we have

$$\sum_{i \in \mathcal{N}} \sum_{t=1}^T \eta_t \left\langle \nabla_i \hat{c}_i(x^t) - \nabla_i c_i\left(x^t\right), \omega_i - x_i^t \right\rangle \leq \sum_{i \in \mathcal{N}} B\ell_i \sum_{t=1}^T \eta_t \sum_{j \in \mathcal{N}} \left(\sigma_{\max}\left(A_j^t\right)^2\right) \mid x^t$$

$$\leq \sum_{i \in \mathcal{N}} B\ell_i \sum_{t=1}^T \frac{1}{\mu(t+1)}$$

$$\leq \frac{B \sum_{i \in \mathcal{N}} \ell_i}{\mu} \sum_{t=1}^T \frac{1}{(t+1)}$$

$$\leq \frac{BL \log(T)}{\mu}$$

where the third inequality is by $\nabla^2 h(x)$ being positive definite, and $\nabla^2 p(x) \geq \mu I$.

Therefore,

$$\sum_{t=1}^T \eta_t \sum_{i \in \mathcal{N}} \left\langle \hat{g}_i^t - \nabla_i c_i\left(x^t\right), \omega_i - x_i^t \right\rangle \leq O\left(\frac{BL \log(T)}{\mu} + \frac{nBd \log^2(1/\delta) \log(T)}{\min\{\sqrt{\mu}, \mu\}}\right).$$

Combining the terms, and with $\eta_t = \frac{1}{2d\sqrt{t}}$, we have

$$\sum_{i \in \mathcal{N}} D_p\left(x_i^*, x_i^{T+1}\right)$$

$$\leq O\left(\frac{nd\nu \log(T)}{\kappa\sqrt{T}} + \frac{nd\zeta B}{T} + \frac{nBL}{\kappa\sqrt{T}} + \frac{ndC_p}{\sqrt{T}} + \frac{nd\log(T)}{\kappa\sqrt{T}} + \frac{dBL\log(T)}{\kappa\mu\sqrt{T}} + \frac{nBd^2\log^2(1/\delta)\log(T)}{\kappa\min\{\sqrt{\mu},\mu\}\sqrt{T}}\right).$$

$\square$

**Lemma E.1.** *With a probability of at least* $1 - \log(T)\delta$, $\delta \leq e^{-1}$, *we have*

$$\sum_{t=1}^{T} \eta_t \left\langle g_i^t - \hat{c}_i^t(x_i^t), \omega_i - x_i^t \right\rangle \leq O\left(\frac{Bd\log^2(1/\delta)\log(T)}{\min\{\sqrt{\mu},\mu\}}\right).$$

*Proof.* Define $Z_t = \eta_t \left\langle g_i^t - \hat{c}_i^t(x_i^t), \omega_i - x_i^t \right\rangle$. $\mathrm{Var}[Z_t] \leq \eta^2 (\omega_i - x_i^t)^\top \mathbb{E}[g_i^t(g_i^t)^\top](\omega_i - x_i^t)$. Then, with $\eta_t = \frac{1}{2d\sqrt{t}}$,

$$\max_t |Z_t| \leq \max_t \left\| \eta_t \left(g_i^t - \hat{c}_i^t(x_i^t)\right)\right\| \left\|\omega_i - x_i^t\right\| \leq O\left(Bd \max_t \|\eta_t(A_i^t)^{-1}z_i^t\|\right) \leq O\left(\max_t \frac{Bd}{\mu(t+1)}\right) \leq O\left(\frac{Bd}{\mu}\right),$$

where the third inequality is by the definition of $A_i^t$.

By the definition of gradient estimator, we have

$$(g_i^t)^\top g_i^t \leq d^2 \left((A_i^t)^{-1}z_i^t\right)^\top \left((A_i^t)^{-1}z_i^t\right) \leq \frac{d^2}{\mu\eta_t(t+1)}.$$

Therefore, with $\eta_t = \frac{1}{2d\sqrt{t}}$

$$(\omega_i - x_i^t)^\top \mathbb{E}[g_i^t(g_i^t)^\top](\omega_i - x_i^t) \leq \frac{d^2\|\omega_i - x_i^t\|^2}{\mu\eta_t(t+1)} \leq \frac{d^2B^2}{\mu\eta_t(t+1)} \leq \frac{dB^2}{\mu\sqrt{t}}.$$

We have

$$\sqrt{\sum_{t=1}^{T} \eta_t^2 (\omega_i - x_i^t)^\top \mathbb{E}[g_i^t(g_i^t)^\top](\omega_i - x_i^t)} \leq \sqrt{\sum_{t=1}^{T} \frac{B^2}{d\mu t^{3/2}}} \leq O\left(\frac{B\sqrt{\log(T)}}{\sqrt{d\mu}}\right).$$

Then, by Lemma 2 of Bartlett et al. (2008), with a probability of at least $1 - \log(T)\delta$, $\delta \leq e^{-1}$,

$$\sum_{t=1}^{T} \eta_t \left\langle g_i^t - \hat{c}_i^t(x_i^t), \omega_i - x_i^t \right\rangle \leq 2\max\left\{2\sqrt{\sum_{t=1}^{T} \mathrm{Var}[Z_t]}, \max_t |Z_t|\log(1/\delta)\right\}$$

$$\leq \max\left\{O\left(\frac{B\sqrt{\log(T)}}{\sqrt{d\mu}}\right), O\left(\frac{Bd\log(1/\delta)}{\mu}\right)\right\} \cdot \log(1/\delta)$$

$$\leq O\left(\frac{Bd\log^2(1/\delta)\log(T)}{\min\{\sqrt{\mu},\mu\}}\right).$$

$\square$

# F    PROOF OF THEOREM 5.4

**Theorem 5.4.** *With $\eta_t = \frac{1}{2d\sqrt{t}}$, $\tau = \frac{1}{T^{1/6}}$, $G_p = \sup_x \|\nabla p(x)\|$ and Algorithm 1, we have*

$$
\mathbb{E}\left[\sum_{i \in \mathcal{N}} \left\langle \nabla_i c_i\left(x^T\right), x_i^T - x_i^*\right\rangle\right]
$$

$$
\leq \tilde{O}\left(\frac{BG_p + \sqrt{d(BL+G)(n\nu + nBL + nd^2)}}{T^{1/6}} + \frac{\sqrt{dBL(BL+G)}}{\sqrt{\mu}T^{1/6}} + \frac{\sqrt{dnC_p(BL+G)}}{\sqrt{\mu}T^{1/4}}\right).
$$

*Proof.* We consider a regularized game with operator $\tilde{F}(x) = [\tilde{F}_i(x)]_{i \in \mathcal{N}}$, where $\tilde{F}_i(x) = \nabla c_i(x) + \tau \nabla p(x_i)$, $\nabla p(x) = [\nabla_i p(x_i)]_{i \in \mathcal{N}}$.

Similar to Lemma J.1, we have

$$
\sum_{i \in \mathcal{N}} D_p\left(x_i^\tau, x_i^{T+1}\right)
$$

$$
\leq O\left(\frac{n\nu \log(T)}{\eta_T \tau T} + \frac{n\mu B}{\eta_T \tau T^{3/2}}\right) + O\left(\frac{nB\sum_{i \in \mathcal{N}}\ell_i}{\tau T^{3/2}} + \frac{n}{\tau T^{3/2}}\right)\frac{\sum_{t=1}^T \eta_t}{\eta_T} + O\left(\frac{nC_p}{\eta_T T}\right)
$$

$$
+ \frac{1}{\eta_T \tau(T+1)}\sum_{i \in \mathcal{N}}\sum_{t=1}^T \eta_t \left\langle \hat{g}_i^t, x_i^t - x_i^{t+1}\right\rangle + \frac{1}{\eta_T \tau(T+1)}\sum_{t=1}^T \eta_t \sum_{i \in \mathcal{M}} \left\langle \hat{g}_i^t - \tilde{F}_i\left(x^t\right), x_i^\tau - x_i^t\right\rangle
$$

$$
+ \frac{1}{\eta_T \tau(T+1)}\sum_{t=1}^T \eta_t \sum_{i \in \mathcal{N}\backslash\mathcal{M}} \left\langle \hat{g}_i^t - \tilde{F}_i\left(x^t\right), \bar{x}_i - x_i^t\right\rangle.
$$

Taking expectation conditioned on $x^t$, we have $\mathbb{E}\left[\|A_i^t \hat{g}_i^t\|^2 \mid x^t\right] = d^2 \mathbb{E}\left[c_i(\hat{x}^t)^2 \|z_i^t\|^2 \mid x^t\right] \leq d^2$. By Lemma J.2, and the choice $\eta_t = \frac{1}{2d\sqrt{t}}$, we have

$$
\sum_{t=1}^T \eta_t \sum_{i \in \mathcal{N}} \mathbb{E}\left[\left\langle \hat{g}_i^t, x_i^t - x_i^{t+1}\right\rangle\right] \leq \sum_{t=1}^T \eta_t^2 \sum_{i \in \mathcal{N}} \mathbb{E}\left[\|A_i^t \hat{g}_i^t\|^2\right] \leq nd^2 \sum_{t=1}^T \eta_t^2.
$$

By Lemma J.5, for any $\omega_i \in \mathcal{X}_i$, we have

$$
\sum_{i \in \mathcal{N}}\sum_{t=1}^T \eta_t \mathbb{E}\left[\left\langle \hat{g}_i^t - \nabla_i c_i\left(x^t\right), \omega_i - x_i^t\right\rangle \mid x^t\right] = \sum_{i \in \mathcal{N}}\sum_{t=1}^T \eta_t \mathbb{E}\left[\left\langle \nabla_i \hat{c}_i(x^t) - \nabla_i c_i\left(x^t\right), \omega_i - x_i^t\right\rangle \mid x^t\right]
$$

$$
\leq \sum_{i \in \mathcal{N}}\sum_{t=1}^T \eta_t \mathbb{E}\left[\|\nabla_i \hat{c}_i(x^t) - \nabla_i c_i\left(x^t\right)\| \|\omega_i - x_i^t\| \mid x^t\right]
$$

$$
\leq \sum_{i \in \mathcal{N}} B\ell_i \sum_{t=1}^T \eta_t \mathbb{E}\left[\sum_{j \in \mathcal{N}}\left(\sigma_{\max}\left(A_j^t\right)^2\right) \mid x^t\right]
$$

$$
\leq \sum_{i \in \mathcal{N}} B\ell_i \sum_{t=1}^T \frac{1}{\mu(t+1)}
$$

$$
\leq \frac{B\sum_{i \in \mathcal{N}}\ell_i}{\mu}\sum_{t=1}^T \frac{1}{(t+1)}.
$$

where the third inequality is by $\nabla^2 h(x)$ being positive definite, and $\nabla^2 p(x) \geq \mu I$.

Combing and rearranging the terms, we have

$$
\mathbb{E}\left[\sum_{i\in\mathcal{N}} D_p\left(x_i^\tau, x_i^{T+1}\right)\right]
$$

$$
\leq O\left(\frac{n\nu\log(T)}{\eta_T\tau T} + \frac{n\zeta B}{\eta_T\tau T^{3/2}}\right) + O\left(\frac{nB\sum_{i\in\mathcal{N}}\ell_i}{\tau\sqrt{T}} + \frac{n}{\tau\sqrt{T}}\right) + O\left(\frac{nC_p}{\eta_T T}\right) + O\left(\frac{nd^2}{\tau\eta_T T}\sum_{t=1}^{T}\eta_t^2 + \frac{B\sum_{i\in\mathcal{N}}\ell_i}{\tau\mu\eta_T T}\sum_{t=1}^{T}\frac{1}{t}\right).
$$

Take $\eta_t = \frac{1}{2d\sqrt{t}}$, we have

$$
\mathbb{E}\left[\sum_{i\in\mathcal{N}} D_p\left(x_i^\tau, x_i^{T+1}\right)\right]
$$

$$
\leq O\left(\frac{nd\nu\log(T)}{\tau\sqrt{T}} + \frac{nd\zeta B}{\tau T} + \frac{nB\sum_{i\in\mathcal{N}}\ell_i}{\tau\sqrt{T}} + \frac{n}{\tau\sqrt{T}} + \frac{ndC_p}{\sqrt{T}} + \frac{nd\log(T)}{\tau\sqrt{T}} + \frac{dB\log(T)\sum_{i\in\mathcal{N}}\ell_i}{\tau\mu\sqrt{T}}\right).
$$

We can decompose as

$$
\left\langle F\left(x^T\right), x^T - x^*\right\rangle
$$
$$
= \left\langle F\left(x^T\right), x^T - x^\tau\right\rangle + \left\langle F\left(x^T\right), x^\tau - x^*\right\rangle
$$
$$
\leq G\left\|x^T - x^\tau\right\| + \left\langle F\left(x^\tau\right) + \tau\nabla p(x^\tau), x^\tau - x^*\right\rangle + \left\langle F\left(x^T\right) - F\left(x^\tau\right), x^\tau - x^*\right\rangle + \tau B\left\|\nabla p(x^\tau)\right\|
$$
$$
\leq \sum_{i\in\mathcal{N}}(B\ell_i + G)\left\|x_i^T - x^\tau\right\| + \tau B\left\|\nabla p(x^\tau)\right\|.
$$

Since $\nabla^2 p(x) \succeq \mu I$, we have $\|x_i^\tau - x_i^T\| \leq \sqrt{D_p(x_i^\tau, x_i^T)}$. Let $G_p = \sup_x \|\nabla p(x)\|$, $L = \sum_{i\in\mathcal{N}}\ell_i$, we have

$$
\mathbb{E}\left[\sum_{i\in\mathcal{N}}\left\langle \nabla_i c_i\left(x^T\right), x_i^T - x_i^*\right\rangle\right]
$$

$$
\leq O\left(\tau B G_p\right) + \tilde{O}\left(\frac{\sqrt{d(BL+G)(n\nu + nBL + nd^2)}}{\sqrt{\tau}T^{1/4}}\right) + \tilde{O}\left(\frac{\sqrt{dBL(BL+G)}}{\sqrt{\tau\mu}T^{1/4}}\right) + O\left(\frac{\sqrt{dnC_p(BL+G)}}{\sqrt{\mu}T^{1/4}}\right)
$$

$$
\leq \tilde{O}\left(\frac{BG_p + \sqrt{d(BL+G)(n\nu + nBL + nd^2)}}{T^{1/6}}\right) + \tilde{O}\left(\frac{\sqrt{dBL(BL+G)}}{\sqrt{\mu}T^{1/6}}\right) + O\left(\frac{\sqrt{dnC_p(BL+G)}}{\sqrt{\mu}T^{1/4}}\right),
$$

where the last inequality is by taking $\tau = \frac{1}{T^{1/6}}$. □

## G  PROOF OF PROPOSITION 5.1

**Proposition 5.1.** *With $\eta_t = \frac{1}{2dt^{3/4}}, \delta_t = \frac{1}{t^{1/4}}$, and suppose every player employ Algorithm 1, we have $\frac{1}{T} \sum_{t=1}^{T} \mathbb{E}\left[\mathrm{SW}(\hat{x})\right] = O\left(\frac{C_1 \mathrm{OPT}}{(1-C_2)} + \frac{n\nu d \log(T)}{(1-C_2)T^{1/4}} + \frac{\sqrt{n}B \sum_{i \in \mathcal{N}} \ell_i}{(1-C_2)T^{1/4}}\right)$.*

*Proof.* By Theorem 5.3, we have

$$\sum_{t=1}^{T} \sum_{i \in \mathcal{N}} \mathbb{E}\left[c_i\left(\hat{x}_i^t, \hat{x}_{-i}^t\right)\right] \leq \sum_{t=1}^{T} \sum_{i \in \mathcal{N}} \mathbb{E}\left[c_i\left(\omega_i, \hat{x}_{-i}^t\right)\right] + O\left(n\nu d T^{3/4} \log(T) + \sqrt{n}B T^{3/4} \sum_{i \in \mathcal{N}} \ell_i\right)$$

$$\leq C_1 \mathrm{OPT} \cdot T + C_2 \sum_{t=1}^{T} \mathbb{E}\left[\mathrm{SW}(\hat{x})\right] + O\left(n\nu d T^{3/4} \log(T) + \sqrt{n}B T^{3/4} \sum_{i \in \mathcal{N}} \ell_i\right).$$

As $\sum_{t=1}^{T} \sum_{i \in \mathcal{N}} \mathbb{E}\left[c_i\left(\hat{x}_i^t, \hat{x}_{-i}^t\right)\right] = \mathbb{E}\left[\mathrm{SW}(\hat{x})\right]$, we solve for $\mathbb{E}\left[\mathrm{SW}(\hat{x})\right]$ and obtain

$$\frac{1}{T} \sum_{t=1}^{T} \mathbb{E}\left[\mathrm{SW}(\hat{x})\right] = O\left(\frac{C_1 \mathrm{OPT}}{(1-C_2)} + \frac{n\nu d \log(T)}{(1-C_2)T^{1/4}} + \frac{\sqrt{n}B \sum_{i \in \mathcal{N}} \ell_i}{(1-C_2)T^{1/4}}\right).$$

$\square$

## H  PROOF OF THEOREM 6.1

**Theorem 6.1.** *With* $\sum_{t=1}^{T} \sum_{i \in \mathcal{N}} \max_x \|\nabla_i c_i(x) - \nabla_i c_i^t(x)\|_2 = T^\alpha$, *take* $\eta_t = \frac{1}{2dt^{3/4}}$, $\delta_t = \frac{1}{t^{1/4}}$, *and under Algorithm 1, we have* $\mathbb{E}\left[\sum_{i \in \mathcal{N}} D_p\left(x_i^*, x_i^{T+1}\right)\right] \leq O\left(\frac{nd\nu \log(T)}{\kappa T^{1/4}} + \frac{n\zeta dB}{T^{3/4}} + \frac{nBL}{\kappa\sqrt{T}} + \frac{ndC_p}{T^{1/4}} + \frac{nd \log(T)}{\kappa T^{1/4}} + \frac{\sqrt{n}B^2 L \log(T)}{\kappa T^{1/4}} + \frac{B}{T^{1/4-\alpha}}\right)$.

*Proof.* Similar to Theorem 5.1, we have

$$\sum_{i \in \mathcal{N}} D_p\left(x_i^*, x_i^{T+1}\right)$$

$$\leq O\left(\frac{n\nu \log(T)}{\eta_T \kappa T} + \frac{n\zeta B}{\eta_T T^{3/2}}\right) + O\left(\frac{nB \sum_{i \in \mathcal{N}} \ell_i}{\kappa T^{3/2}} + \frac{n}{\kappa T^{3/2}}\right)\frac{\sum_{t=1}^{T} \eta_t}{\eta_T} + O\left(\frac{nC_p}{\eta_T T}\right)$$

$$+ \frac{1}{\kappa\eta_T(T+1)} \sum_{i \in \mathcal{N}} \sum_{t=1}^{T} \eta_t \langle \hat{g}_i^t, x_i^t - x_i^{t+1}\rangle + \frac{1}{\kappa\eta_T(T+1)} \sum_{t=1}^{T} \eta_t \sum_{i \in \mathcal{M}} \langle \hat{g}_i^t - \nabla_i c_i^t\left(x^t\right), x_i^* - x_i^t\rangle$$

$$+ \frac{1}{\kappa\eta_T(T+1)} \sum_{t=1}^{T} \eta_t \sum_{i \in \mathcal{N}\setminus\mathcal{M}} \langle \hat{g}_i^t - \nabla_i c_i^t\left(x^t\right), \bar{x}_i - x_i^t\rangle + B \sum_{t=1}^{T} \Delta^t,$$

where $\Delta^t = \sum_{i \in \mathcal{N}} \max_x \|\nabla_i c_i(x) - \nabla_i c_i^t(x)\|_2$.

We now upper bound the remaining terms by discussing them by cases.

When $\mu = 0$, taking expectation conditioned on $x^t$, we have $\mathbb{E}\left[\|A_i^t \hat{g}_i^t\|^2 \mid x^t\right] = \frac{d^2}{\delta_t^2} \mathbb{E}\left[c_i^t(\hat{x}^t)^2 \|z_i^t\|^2 \mid x^t\right] \leq \frac{d^2}{\delta_t^2}$. By Lemma J.2, and the choice $\eta_t = \frac{1}{2d\sqrt{t}}$, we have

$$\sum_{t=1}^{T} \eta_t \sum_{i \in \mathcal{N}} \mathbb{E}\left[\langle \hat{g}_i^t, x_i^t - x_i^{t+1}\rangle\right] \leq \sum_{t=1}^{T} \eta_t^2 \sum_{i \in \mathcal{N}} \mathbb{E}\left[\|A_i^t \hat{g}_i^t\|^2\right] \leq nd^2 \sum_{t=1}^{T} \frac{\eta_t^2}{\delta_t^2}.$$

By the definition of $\hat{c}_i$,

$$\sum_{i \in \mathcal{N}} \sum_{t=1}^{T} \eta_t \mathbb{E}\left[\langle \hat{g}_i^t - \nabla_i c_i^t\left(x^t\right), \omega_i - x_i^t\rangle \mid x^t\right]$$

$$= \sum_{i \in \mathcal{N}} \sum_{t=1}^{T} \eta_t \mathbb{E}\left[\langle \nabla_i \hat{c}_i^t(x^t) - \nabla_i c_i^t\left(x^t\right), \omega_i - x_i^t\rangle \mid x^t\right]$$

$$= \sum_{i \in \mathcal{N}} \sum_{t=1}^{T} \eta_t \mathbb{E}\left[\mathbb{E}_{w_i \sim \mathbb{B}^d} \mathbb{E}_{\mathbf{z}_{-i} \sim \Pi_{j\neq i}\mathbb{S}^d} \langle \nabla_i c_i^t\left(x_i^t + \delta_t A_i^t w_i, \hat{x}_{-i}^t\right) - \nabla_i c_i^t\left(x^t\right), \omega_i - x_i^t\rangle \mid x^t\right]$$

$$\leq B \sum_{i \in \mathcal{N}} \sum_{t=1}^{T} \eta_t \mathbb{E}\left[\mathbb{E}_{w_i \sim \mathbb{B}^d} \mathbb{E}_{\mathbf{z}_{-i} \sim \Pi_{j\neq i}\mathbb{S}^d} \|\nabla_i c_i^t\left(x_i^t + \delta_t A_i^t w_i, \hat{x}_{-i}^t\right) - \nabla_i c_i^t\left(x^t\right)\| \mid x^t\right]$$

By the smoothness of $c_i^t$,

$$\mathbb{E}_{w_i \sim \mathbb{B}^d} \mathbb{E}_{\mathbf{z}_{-i} \sim \Pi_{j\neq i}\mathbb{S}^d}\left[\|\nabla_i c_i^t\left(x_i^t + \delta_t A_i^t w_i, \hat{x}_{-i}^t\right) - \nabla_i c_i^t\left(x^t\right)\|\right]$$

$$\leq \ell_i \mathbb{E}_{w_i \sim \mathbb{B}^d} \mathbb{E}_{\mathbf{z}_{-i} \sim \Pi_{j\neq i}\mathbb{S}^d}\left[\sqrt{\delta_t^2 \|A_i w_i\|^2 + \delta_t^2 \sum_{j\neq i} \|A_j z_j\|^2}\right].$$

Since $p$ is convex, $\nabla^2 p(x)$ is positive semi-definite, and $A_i^t \preceq (\nabla^2 h(x_i))^{-1/2}$. For $\bar{x}_i^t = x_i^t + A_i^t w_i^t$. Define $\|v\|_x = \sqrt{v^\top \nabla^2 h(x) v}$, we have $\|\bar{x}_i^t - x_i^t\|_{x_i} \leq \|\omega_i^t\| \leq 1$, and $\bar{x}_i^t \in W(x_i^t)$, where $W(x_i) = \{x_i' \in \mathbb{R}^d, \|x_i' - x_i\|_{x_i} \leq 1\}$ is the Dikin ellipsoid. Since $W(x_i) \subseteq \mathcal{X}_i, \forall x_i \in int(\mathcal{X}_i)$, we can upper

bound $\|A_i w_i\|^2$ by $B^2$, the diameter of the set $\mathcal{X}_i$. Hence $\|\nabla_i \hat{c}_i(x^t) - \nabla_i c_i(x^t)\| \le \ell_i \delta_t \sqrt{n} B$. By Lemma J.5

$$\sum_{i \in \mathcal{N}} \sum_{t=1}^{T} \eta_t \mathbb{E}\left[\langle \hat{g}_i^t - \nabla_i c_i^t(x^t), \omega_i - x_i^t \rangle \mid x^t\right] = \sum_{i \in \mathcal{N}} \sum_{t=1}^{T} \eta_t \mathbb{E}\left[\langle \nabla_i \hat{c}_i^t(x^t) - \nabla_i c_i^t(x^t), \omega_i - x_i^t \rangle \mid x^t\right]$$

$$\le \sum_{i \in \mathcal{N}} \sum_{t=1}^{T} \eta_t \mathbb{E}\left[\|\nabla_i \hat{c}_i^t(x^t) - \nabla_i c_i^t(x^t)\| \|\omega_i - x_i^t\| \mid x^t\right]$$

$$\le \sqrt{n} B^2 \sum_{i \in \mathcal{N}} \ell_i \sum_{t=1}^{T} \eta_t \delta_t.$$

Let $L = \sum_{i \in \mathcal{N}} \ell_i$. When $\mu = 0$, combing and rearranging the terms, we have

$$\mathbb{E}\left[\sum_{i \in \mathcal{N}} D_p(x_i^*, x_i^{T+1})\right]$$

$$\le O\left(\frac{n\nu \log(T)}{\kappa \eta_T T} + \frac{n\zeta B}{\eta_T T^{3/2}} + \frac{nBL}{\kappa \sqrt{T}} + \frac{n}{\kappa \sqrt{T}} + \frac{nC_p}{\eta_T T} + \frac{nd^2}{\kappa \eta_T T} \sum_{t=1}^{T} \frac{\eta_t^2}{\delta_t^2} + \frac{\sqrt{n} B^2 L \sum_{t=1}^{T} \eta_t \delta_t}{\kappa \eta_T T} + \frac{B \sum_{t=1}^{T} \Delta^t}{\eta_T T}\right).$$

Take $\eta_t = \frac{1}{2dt^{3/4}}$, $\delta_t = \frac{1}{t^{1/4}}$, then $\sum_{t=1}^{T} \frac{\eta_t^2}{\delta_t^2} = O\left(\sum_{t=1}^{T} \frac{1}{t}\right) = O(\log(T))$, and $\sum_{t=1}^{T} \eta_t \delta_t = O\left(\sum_{t=1}^{T} \frac{1}{t}\right) = O(\log(T))$. Hence, we have

$$\mathbb{E}\left[\sum_{i \in \mathcal{N}} D_p(x_i^*, x_i^{T+1})\right] \le O\left(\frac{nd\nu \log(T)}{\kappa T^{1/4}} + \frac{n\zeta dB}{T^{3/4}} + \frac{nBL}{\kappa \sqrt{T}} + \frac{ndC_p}{T^{1/4}} + \frac{nd \log(T)}{\kappa T^{1/4}} + \frac{\sqrt{n} B^2 L \log(T)}{\kappa T^{1/4}} + \frac{B\Delta}{T^{1/4}}\right),$$

where $\Delta = \sum_{t=1}^{T} \sum_{i \in \mathcal{N}} \max_x \|\nabla_i c_i(x) - \nabla_i c_i^t(x)\|_2$. $\qquad\square$

## I   PROOF OF THEOREM 6.2

**Theorem 6.2.** *Assume $V_i(T) \leq T^\varphi$, $\varphi \in [0,1]$. Take $\eta_t = \frac{1}{2dt^{\frac{(1-\varphi)}{3}}}$, $\delta_t = \frac{1}{t^{1/2}}$, and under Algorithm 1, we have $\frac{1}{T}\sum_{t=1}^{T}\sum_{i\in\mathcal{N}}\left\langle\nabla_i c_i^t\left(\hat{x}_i^t,\hat{x}_{-i}^t\right),\hat{x}_i^t-x_i^{t,*}\right\rangle = \tilde{O}\left(\frac{n\nu d + Ln^{3/2}B^2 + nG}{T^{\frac{2(1-\varphi)}{3}}} + \frac{n}{T^{\frac{9}{8}-\frac{(4\varphi+5)^2}{72}}}\right)$.*

*Proof.* We first fix a player $i$ decomposes

$$\sum_{t=1}^{T}\left\langle\nabla_i c_i^t\left(\hat{x}_i^t,\hat{x}_{-i}^t\right),\hat{x}_i^t-x_i^{t,*}\right\rangle = \sum_{t=1}^{T}\left\langle\nabla_i c_i^t\left(\hat{x}_i^t,\hat{x}_{-i}^t\right),\hat{x}_i^t-\omega_i\right\rangle + \sum_{t=1}^{T}\left\langle\nabla_i c_i^t\left(\hat{x}_i^t,\hat{x}_{-i}^t\right),\omega_i-x_i^{t,*}\right\rangle.$$

For the second term, we partition the horizon of play $T$ into $m$ batches $T_k$, $k\in[m]$, each of length $|T_k| = T^q$, $q \in [0,1]$. We will determine $q$ later. Note that the number of batches is thus $m = T^{1-q}$. For the batch $T_k$, we pick $\omega_i$ to be the Nash equilibrium of the first game. Then

$$\sum_{t\in[T_k]}\left\langle\nabla_i c_i^t\left(\hat{x}_i^t,\hat{x}_{-i}^t\right),\omega_i-x_i^{t,*}\right\rangle \leq \sum_{t\in[T_k]}\left\|\nabla_i c_i^t\left(\hat{x}_i^t,\hat{x}_{-i}^t\right)\right\|\left\|\omega_i-x_i^{t,*}\right\|$$

$$\leq GT^q\max_{t\in[T_k]}\left\|\omega_i-x_i^{t,*}\right\|$$

$$\leq GT^q\sum_{t\in[T_k]}\left\|x_i^{t+1,*}-x_i^{t,*}\right\|$$

$$\leq GT^q V_i(T_k),$$

where the third inequality is by the definition of $\omega_i$.

Therefore, we have

$$\sum_{t=1}^{T}\left\langle\nabla_i c_i^t\left(\hat{x}_i^t,\hat{x}_{-i}^t\right),\hat{x}_i^t-x_i^{t,*}\right\rangle = \sum_{k=1}^{m}\sum_{t\in[T_k]}\left\langle\nabla_i c_i^t\left(\hat{x}_i^t,\hat{x}_{-i}^t\right),\hat{x}_i^t-\omega_i\right\rangle + GT^q V_i(T).$$

Define the smoothed version of $c_i$ as $\hat{c}_i^t(x) = \mathbb{E}_{w_i\sim\mathbb{B}^d}\left[c_i^t\left(x_i+\delta A_i w_i, x_{-i}\right)\right]$. Then, for batch $T_k$, we decompose $\sum_{t=1}^{T}\left\langle\nabla_i c_i\left(\hat{x}_i^t,\hat{x}_{-i}^t\right),\hat{x}_i^t-\omega_i\right\rangle$ as

$$\sum_{t\in[T_k]}\left\langle\nabla_i c_i\left(\hat{x}_i^t,\hat{x}_{-i}^t\right),\hat{x}_i^t-\omega_i\right\rangle$$

$$= \sum_{t\in[T_k]}\left\langle\nabla_i\hat{c}_i\left(\hat{x}_i^t,\hat{x}_{-i}^t\right),\hat{x}_i^t-\omega_i\right\rangle + \sum_{t\in[T_k]}\left\langle\nabla_i c_i\left(\hat{x}_i^t,\hat{x}_{-i}^t\right)-\nabla_i\hat{c}_i\left(\hat{x}_i^t,\hat{x}_{-i}^t\right),\hat{x}_i^t-\omega_i\right\rangle$$

$$\leq \sum_{t\in[T_k]}\left\langle\nabla_i\hat{c}_i\left(\hat{x}_i^t,\hat{x}_{-i}^t\right),\hat{x}_i^t-\omega_i\right\rangle + B\sum_{t\in[T_k]}\left\|\nabla_i c_i\left(\hat{x}_i^t,\hat{x}_{-i}^t\right)-\nabla_i\hat{c}_i\left(\hat{x}_i^t,\hat{x}_{-i}^t\right)\right\|_2.$$

For the first term, recall that by the update rule, we have,

$$D_h\left(\omega_i,\hat{x}_i^{t+1}\right) = D_h\left(\omega_i,\hat{x}_i^t\right) + \eta_t\left\langle\nabla\hat{c}_i^t\left(\hat{x}^t\right),\omega_i-\hat{x}_i^t\right\rangle + \eta_t\left\langle\hat{g}_i^t-\nabla\hat{c}_i^t\left(\hat{x}^t\right),\omega_i-\hat{x}_i^t\right\rangle$$
$$+ \eta_t\left\langle\hat{g}_i^t,\hat{x}_i^t-\hat{x}_i^{t+1}\right\rangle.$$

By Lemma J.5, for any $\omega_i \in \mathcal{X}_i$, we have

$$\mathbb{E}\left[\left\langle\hat{g}_i^t-\nabla\hat{c}_i^t\left(\hat{x}^t\right),\omega_i-\hat{x}_i^t\right\rangle\mid\hat{x}^t\right] = \mathbb{E}\left[\left\langle\nabla_i\hat{c}_i^t(\hat{x}^t)-\nabla_i\hat{c}_i^t\left(\hat{x}^t\right),\omega_i-\hat{x}_i^t\right\rangle\mid\hat{x}^t\right] = 0.$$

Therefore,

$$\mathbb{E}\left[D_h\left(\omega_i,\hat{x}_i^{t+1}\right)\right] = \mathbb{E}\left[D_h\left(\omega_i,\hat{x}_i^t\right) + \eta_t\left\langle\nabla\hat{c}_i^t\left(\hat{x}^t\right),\omega_i-\hat{x}_i^t\right\rangle\right] + \eta_t\mathbb{E}\left[\left\langle\hat{g}_i^t,\hat{x}_i^t-\hat{x}_i^{t+1}\right\rangle\right].$$

Rearranging the terms yields

$$\mathbb{E}\left[\langle \nabla \hat{c}_i^t(\hat{x}^t), \hat{x}_i^t - \omega_i\rangle\right] \leq \mathbb{E}\left[\frac{(D_h(\omega_i, \hat{x}_i^t) - D_h(\omega_i, \hat{x}_i^{t+1}))}{\eta_t} + \eta_t\langle \hat{g}_i^t, \hat{x}_i^t - \hat{x}_i^{t+1}\rangle\right].$$

By Lemma J.2, we have $\mathbb{E}\left[\langle \hat{g}_i^t, \hat{x}_i^t - \hat{x}_i^{t+1}\rangle\right] \leq \eta_t\mathbb{E}\left[\|A_i^t\hat{g}_i^t\|^2\right]$. Taking expectation conditioned on $\hat{x}^t$, we have $\mathbb{E}\left[\|A_i^t\hat{g}_i^t\|^2 \mid \hat{x}^t\right] = \frac{d^2}{\delta_t^2}\mathbb{E}\left[\tilde{c}_i^t(\hat{x}^t)^2\|z_i^t\|^2 \mid \hat{x}^t\right] \leq \frac{d^2}{\delta_t^2}$, and therefore $\mathbb{E}\left[\langle \hat{g}_i^t, \hat{x}_i^t - \hat{x}_i^{t+1}\rangle\right] \leq \frac{\eta_t d^2}{\delta_t^2}$.

Taking summation over $T$, and take $\eta_t = \frac{1}{2dt^p}$, $\delta_t = \frac{1}{t^r}$ we have

$$\sum_{t\in[T_k]}\mathbb{E}\left[\langle \nabla \hat{c}_i^t(\hat{x}^t), \hat{x}_i^t - \omega_i\rangle\right] \leq dT^p\mathbb{E}\left[D_h(\omega_i, x_i^1)\right] + \sum_{t\in[T_k]}\frac{\eta_t d^2}{\delta^2}$$

$$\leq O\left(dT^p\mathbb{E}\left[D_h(\omega_i, x_i^1)\right] + T^{q(p-2r)}\right),$$

as we assumed $D_p(x_i, x_i')$ is bounded for any $x_i, x_i'$.

Define $\pi_x(y) = \inf\left\{t \geq 0 : x + \frac{1}{t}(y-x) \in \mathcal{X}_i\right\}$. Notice that $x_i^1(x) = \arg\min_{x_i\in\mathcal{X}_i} h(x_i)$, so $D_h(\omega_i, x_i^1) = h(\omega_i) - h(x_i^1)$.

- If $\pi_{x_i^1}(\omega_i) \leq 1 - \frac{1}{\sqrt{T^q}}$, then by Lemma J.6, $D_h(\omega_i, x_i^1) = \nu\log(T^q)$, and $\sum_{t=1}^{T}\mathbb{E}\left[\hat{c}_i(\hat{x}_i^t, x_{-i}^t) - \hat{c}_i(\omega_i, x_{-i}^t)\right] = O\left(\nu dT^{1-p}\log(T^q)\right)$.

- Otherwise, we find a point $\omega_i'$ such that $\|\omega_i' - \omega_i\| = O(1/\sqrt{T^q})$ and $\pi_{x_i^1}(\omega_i') \leq 1 - \frac{1}{\sqrt{T^q}}$. Then $D_h(\omega_i', x_i^1) = \nu\log(T^q)$,

$$\langle \nabla_i\hat{c}_i^t(\omega_i', x_{-i}^t), \omega_i' - \omega_i\rangle \leq \|\nabla_i\hat{c}_i^t(\omega_i', x_{-i}^t)\|\|\omega_i' - \omega_i\| \leq \frac{G}{\sqrt{T^q}}.$$

  Therefore, $\sum_{t\in[T_k]}\mathbb{E}\left[\hat{c}_i(\hat{x}_i^t, x_{-i}^t) - \hat{c}_i(\omega_i, x_{-i}^t)\right] = O\left(\nu dT^p\log(T^q) + GT^{q/2} + T^{q(p-2r)}\right)$.

By the definition of $\hat{c}_i$ and the smoothness of $c_i$,

$$\|\nabla_i\hat{c}_i(\hat{x}^t) - \nabla_i c_i(\hat{x}^t)\| = \left\|\mathbb{E}_{w_i\sim\mathbb{B}^d}\mathbb{E}_{\mathbf{z}_{-i}\sim\Pi_{j\neq i}\mathbb{S}^d}\left[\nabla_i c_i(\hat{x}_i^t + \delta_t A_i^t w_i, \hat{x}_{-i}^t) - \nabla_i c_i(\hat{x}^t)\right]\right\|$$

$$\leq \ell_i\sqrt{\mathbb{E}_{w_i\sim\mathbb{B}^d}\mathbb{E}_{\mathbf{z}_{-i}\sim\Pi_{j\neq i}\mathbb{S}^d}\left[\delta_t^2\|\delta_t A_i w_i\|^2 + \delta_t^2\sum_{j\neq i}\|A_j z_j\|^2\right]}.$$

Since $p$ is convex, $\nabla^2 p(x)$ is positive semi-definite, and $A_i^t \preceq (\nabla^2 h(x_i))^{-1/2}$. For $\bar{x}_i^t = \hat{x}_i^t + A_i^t w_i^t$. Define $\|v\|_x = \sqrt{v^\top\nabla^2 h(x)v}$, we have $\|\bar{x}_i^t - \hat{x}_i^t\|_{x_i} \leq \|\omega_i^t\| \leq 1$, and $\bar{x}_i^t \in W(\hat{x}_i^t)$, where $W(x) = \{x_i' \in \mathbb{R}^d, \|x_i' - x_i\|_{x_i} \leq 1\}$ is the Dikin ellipsoid. Since $W(x_i) \subseteq \mathcal{X}_i, \forall x_i \in int(\mathcal{X}_i)$, we can upper bound $\|A_i w_i\|^2$ by $B^2$, the diameter of the set $\mathcal{X}_i$. Hence $\|\nabla_i\hat{c}_i(\hat{x}^t) - \nabla_i c_i(\hat{x}^t)\| \leq \ell_i\delta_t\sqrt{n}B$.

With $\delta_t = \frac{1}{t^r}$, we have

$$\sum_{t\in[T_k]}\mathbb{E}\left[\langle \nabla_i c_i(\hat{x}_i^t, \hat{x}_{-i}^t), \hat{x}_i^t - \omega_i\rangle\right] = O\left(\nu dT^p\log(T^q) + GT^{q/2} + T^{q(p-2r)} + \ell_i\sqrt{n}B^2 T^{q(1-r)}\right).$$

Combining, as $m = T^{1-q}$ we have

$$\sum_{t=1}^{T}\mathbb{E}\left[\langle \nabla_i c_i^t(\hat{x}_i^t, \hat{x}_{-i}^t), \hat{x}_i^t - x_i^{t,*}\rangle\right]$$

$$= O\left(GT^q V_i(T)\right) + \sum_{j\in[m]}\tilde{O}\left(\nu dT^{1-p} + GT^{q/2} + T^{q(p-2r)} + \ell_i\sqrt{n}B^2 T^{q(1-r)}\right)$$

$$= \tilde{O}\left(\nu dT^{(1-q)+p} + GT^{(1-q)+q/2} + T^{(1-q)+q(p-2r)} + \ell_i\sqrt{n}B^2 T^{(1-q)+q(1-r)} + GT^q V_i(T)\right).$$

When $V_i(T) = T^\varphi$, $\varphi \in [0,1]$, we set $q = \frac{2(1-\varphi)}{3}$, $p = \frac{(1-\varphi)}{3}$, $r = \frac{1}{2}$, we have

$$\sum_{t=1}^{T} \mathbb{E}\left[\langle \nabla_i c_i^t\left(\hat{x}_i^t, \hat{x}_{-i}^t\right), \hat{x}_i^t - x_i^{t,*}\rangle\right] = \tilde{O}\left(\left(\nu d + G + \ell_i \sqrt{n} B^2\right) T^{\frac{1+2\varphi}{3}} + T^{\frac{(2\varphi+1)(\varphi+2)}{9}}\right).$$

Divided by $T$, we have

$$\frac{1}{T} \sum_{t=1}^{T} \mathbb{E}\left[\langle \nabla_i c_i^t\left(\hat{x}_i^t, \hat{x}_{-i}^t\right), \hat{x}_i^t - x_i^{t,*}\rangle\right] = \tilde{O}\left(\frac{\nu d + G + \ell_i \sqrt{n} B^2}{T^{\frac{2(1-\varphi)}{3}}} + \frac{1}{T^{\frac{9}{8} - \frac{(4\varphi+5)^2}{72}}}\right).$$

Sum over $i \in \mathcal{N}$ and we have the claimed result. $\qquad\square$

## J  Auxiliary Lemmas

**Lemma J.1.** *With the update rule equation 1,*

$$\sum_{i \in \mathcal{N}} D_p \left( x_i^*, x_i^{T+1} \right)$$

$$\leq O \left( \frac{n\nu \log(T)}{\eta_T \kappa T} + \frac{n\zeta B}{\eta_T T^{3/2}} \right) + O \left( \frac{nB \sum_{i \in \mathcal{N}} \ell_i}{\kappa T^{3/2}} + \frac{n}{\kappa T^{3/2}} \right) \frac{\sum_{t=1}^{T} \eta_t}{\eta_T} + O \left( \frac{nC_p}{\eta_T T} \right)$$

$$+ \frac{1}{\kappa \eta_T (T+1)} \sum_{i \in \mathcal{N}} \sum_{t=1}^{T} \eta_t \left\langle \hat{g}_i^t, x_i^t - x_i^{t+1} \right\rangle + \frac{1}{\kappa \eta_T (T+1)} \sum_{t=1}^{T} \eta_t \sum_{i \in \mathcal{M}} \left\langle \hat{g}_i^t - \nabla_i c_i \left( x^t \right), x_i^* - x_i^t \right\rangle$$

$$+ \frac{1}{\kappa \eta_T (T+1)} \sum_{t=1}^{T} \eta_t \sum_{i \in \mathcal{N} \setminus \mathcal{M}} \left\langle \hat{g}_i^t - \nabla_i c_i \left( x^t \right), \bar{x}_i - x_i^t \right\rangle,$$

*where $\bar{x}_i$ is a point such that $\|\bar{x}_i - x_i^*\| = O(1/\sqrt{T})$ and $\inf \left\{ t \geq 0 : x_i^1 + \frac{1}{t}(\bar{x}_i - x_i^1) \in \mathcal{X}_i \right\} \leq 1 - 1/\sqrt{T}$.*

*Proof.* By the update rule equation 1, we have

$$\eta_t \hat{g}_i^t + \eta_t \kappa(t+1) \left( \nabla p \left( x_i^{t+1} \right) - \nabla p \left( x_i^t \right) \right) + \left( \nabla h \left( x_i^{t+1} \right) - \nabla h \left( x_i^t \right) \right) = 0.$$

For a fixed point $\omega_i$, by the three-point equality of Bregman divergence, we have

$$D_h \left( \omega_i, x_i^{t+1} \right)$$

$$= D_h \left( \omega_i, x_i^t \right) - D_h \left( x_i^{t+1}, x_i^t \right) + \left\langle \nabla h \left( x_i^t \right) - \nabla h \left( x_i^{t+1} \right), \omega_i - x_i^{t+1} \right\rangle$$

$$= D_h \left( \omega_i, x_i^t \right) - D_h \left( x_i^{t+1}, x_i^t \right) + \eta_t \left\langle \hat{g}_i^t, \omega_i - x_i^{t+1} \right\rangle + \eta_t \kappa(t+1) \left\langle \nabla p \left( x_i^{t+1} \right) - \nabla p \left( x_i^t \right), \omega_i - x_i^{t+1} \right\rangle$$

$$= D_h \left( \omega_i, x_i^t \right) - D_h \left( x_i^{t+1}, x_i^t \right) + \eta_t \left\langle \hat{g}_i^t, \omega_i - x_i^{t+1} \right\rangle + \eta_t \kappa(t+1) \left( D_p \left( \omega_i, x_i^t \right) - D_p \left( \omega_i, x_i^{t+1} \right) - D_p \left( x_i^{t+1}, x_i^t \right) \right).$$

Rearranging and by the non-negativity of Bregman divergence, we have,

$$D_h \left( \omega_i, x_i^{t+1} \right) + \eta_t \kappa(t+1) D_p \left( \omega_i, x_i^{t+1} \right)$$

$$\leq D_h \left( \omega_i, x_i^t \right) + \eta_t \kappa(t+1) D_p \left( \omega_i, x_i^t \right) + \eta_t \left\langle \hat{g}_i^t, \omega_i - x_i^t \right\rangle + \eta_t \left\langle \hat{g}_i^t, x_i^t - x_i^{t+1} \right\rangle$$

$$= D_h \left( \omega_i, x_i^t \right) + \eta_t \kappa(t+1) D_p \left( \omega_i, x_i^t \right) + \eta_t \left\langle \nabla_i c_i \left( x^t \right), \omega_i - x_i^t \right\rangle + \eta_t \left\langle \hat{g}_i^t - \nabla_i c_i \left( x^t \right), \omega_i - x_i^t \right\rangle + \eta_t \left\langle \hat{g}_i^t, x_i^t - x_i^{t+1} \right\rangle.$$

By Lemma J.3 and the assumption that $c_i(x) - \kappa p(x_i)$ is convex, we have

$$\eta_t \sum_{i \in \mathcal{N}} \left\langle \nabla_i c_i \left( x^t \right), \omega_i - x_i^t \right\rangle \leq -\eta_t \kappa \sum_{i \in \mathcal{N}} \left( D_p \left( x_i^t, \omega_i \right) + D_p \left( \omega_i, x_i^t \right) \right) + \eta_t \sum_{i \in \mathcal{N}} \left\langle \nabla_i c_i \left( \omega \right), \omega_i - x_i^t \right\rangle.$$

Therefore,

$$\sum_{i \in \mathcal{N}} D_h \left( \omega_i, x_i^{t+1} \right) + \eta_t \kappa(t+1) \sum_{i \in \mathcal{N}} D_p \left( \omega_i, x_i^{t+1} \right)$$

$$\leq \sum_{i \in \mathcal{N}} D_h \left( \omega_i, x_i^t \right) + \eta_t \kappa t \sum_{i \in \mathcal{N}} D_p \left( \omega_i, x_i^t \right) + \eta_t \sum_{i \in \mathcal{N}} \left\langle \nabla_i c_i(\omega), \omega_i - x_i^t \right\rangle + \eta_t \sum_{i \in \mathcal{N}} \left\langle \hat{g}_i^t - \nabla_i c_i \left( x^t \right), \omega_i - x_i^t \right\rangle$$

$$+ \eta_t \sum_{i \in \mathcal{N}} \left\langle \hat{g}_i^t, x_i^t - x_i^{t+1} \right\rangle.$$

Summing over $T$, by the non-negativity of Bregman divergence, we have

$$\eta_T \kappa(T+1) \sum_{i \in \mathcal{N}} D_p \left( \omega_i, x_i^{T+1} \right)$$

$$\leq \sum_{i \in \mathcal{N}} D_h \left( \omega_i, x_i^1 \right) + \kappa \sum_{i \in \mathcal{N}} D_p \left( \omega_i, x_i^1 \right) + \sum_{t=1}^{T} \sum_{i \in \mathcal{N}} \eta_t \left\langle \nabla_i c_i(\omega), \omega_i - x_i^t \right\rangle + \sum_{t=1}^{T} \sum_{i \in \mathcal{N}} \eta_t \left\langle \hat{g}_i^t - \nabla_i c_i \left( x^t \right), \omega_i - x_i^t \right\rangle$$

$$+ \sum_{t=1}^{T} \sum_{i \in \mathcal{N}} \eta_t \left\langle \hat{g}_i^t, x_i^t - x_i^{t+1} \right\rangle.$$

Define $\pi_x(y) = \inf \left\{ t \geq 0 : x + \frac{1}{t}(y - x) \in \mathcal{X}_i \right\}$, let us consider $x_i^*$, the equilibrium of the game.

- If $\pi_{x_i^1}(x_i^*) \leq 1 - 1/\sqrt{T}$, we set $\omega_i = x_i^*$. Let this set of player be $\mathcal{M}$

- Otherwise, we find $\bar{x}_i \in \mathcal{X}_i$ such that $\|\bar{x}_i - x_i^*\| = O(1/\sqrt{T})$ and $\pi_{x_i^1}(\bar{x}_i) \leq 1 - 1/\sqrt{T}$. We set $\omega_i = \bar{x}_i$.

By Lemma J.6, and initializing $x_i^1$ to minimize $h$, thus $D_h(\omega_i, x_i^1) = h(\omega_i) - h(x_i^1) \leq \nu \log(T)$. Therefore, we have

$$\eta_T \kappa (T+1) \left( \sum_{i \in \mathcal{M}} D_p \left( x_i^*, x_i^{T+1} \right) + \sum_{i \in \mathcal{N} \setminus \mathcal{M}} D_p \left( \bar{x}_i, x_i^{T+1} \right) \right)$$

$$\leq n\nu \log(T) + \kappa \sum_{i \in \mathcal{M}} D_p \left( x_i^*, x_i^1 \right) + \kappa \sum_{i \in \mathcal{N} \setminus \mathcal{M}} D_p \left( \bar{x}_i, x_i^1 \right) + \sum_{t=1}^{T} \eta_t \sum_{i \in \mathcal{M}} \left\langle \nabla_i c_i(x_{\mathcal{M}}^*, \bar{x}_{\mathcal{N} \setminus \mathcal{M}}), x_i^* - x_i^t \right\rangle$$

$$+ \sum_{t=1}^{T} \eta_t \sum_{i \in \mathcal{N} \setminus \mathcal{M}} \left\langle \nabla_i c_i(x_{\mathcal{M}}^*, \bar{x}_{\mathcal{N} \setminus \mathcal{M}}), \bar{x}_i - x_i^t \right\rangle + \eta_t \sum_{t=1}^{T} \sum_{i \in \mathcal{M}} \left\langle \hat{g}_i^t - \nabla_i c_i \left( x^t \right), x_i^* - x_i^t \right\rangle$$

$$+ \eta_t \sum_{t=1}^{T} \sum_{i \in \mathcal{N} \setminus \mathcal{M}} \left\langle \hat{g}_i^t - \nabla_i c_i \left( x^t \right), \bar{x}_i - x_i^t \right\rangle + \sum_{i \in \mathcal{N}} \sum_{t=1}^{T} \eta_t \left\langle \hat{g}_i^t, x_i^t - x_i^{t+1} \right\rangle.$$

By the three-point inequality and the non-negativity of Bregman divergence, we have

$$\sum_{i \in \mathcal{N} \setminus \mathcal{M}} D_p \left( \bar{x}_i, x_i^{T+1} \right) = \sum_{i \in \mathcal{N} \setminus \mathcal{M}} D_p \left( \bar{x}_i, x_i^* \right) + \sum_{i \in \mathcal{N} \setminus \mathcal{M}} D_p \left( x_i^*, x_i^{T+1} \right) - \sum_{i \in \mathcal{N} \setminus \mathcal{M}} \left\langle \bar{x}_i - x_i^*, \nabla p \left( x_i^{T+1} \right) - \nabla p \left( \bar{x}_i \right) \right\rangle$$

$$\geq \sum_{i \in \mathcal{N} \setminus \mathcal{M}} D_p \left( x_i^*, x_i^{T+1} \right) - \sum_{i \in \mathcal{N} \setminus \mathcal{M}} \left\langle \bar{x}_i - x_i^*, \nabla p \left( x_i^{T+1} \right) - \nabla p \left( \bar{x}_i \right) \right\rangle.$$

By Cauchy-Schwarz and the smoothness of $p$, we have

$$\sum_{i \in \mathcal{N} \setminus \mathcal{M}} \left\langle \bar{x}_i - x_i^*, \nabla p \left( x_i^{T+1} \right) - \nabla p \left( \bar{x}_i \right) \right\rangle \leq \sum_{i \in \mathcal{N} \setminus \mathcal{M}} \| \bar{x}_i - x_i^* \| \left\| \nabla p \left( x_i^{T+1} \right) - \nabla p \left( \bar{x}_i \right) \right\|$$

$$\leq \zeta \sum_{i \in \mathcal{N} \setminus \mathcal{M}} \| \bar{x}_i - x_i^* \| \left\| x_i^{T+1} - \bar{x}_i \right\|$$

$$\leq O \left( \frac{n \zeta B}{\sqrt{T}} \right)$$

As $x_i^*$ is a Nash equilibrium, we have $\sum_{i \in \mathcal{N}} \left\langle \nabla_i c_i(x^*), x_i^* - x_i^t \right\rangle = 0$, therefore,

$$\eta_t \sum_{i \in \mathcal{M}} \left\langle \nabla_i c_i(x_{\mathcal{M}}^*, \bar{x}_{\mathcal{N} \setminus \mathcal{M}}), x_i^* - x_i^t \right\rangle + \eta_t \sum_{i \in \mathcal{N} \setminus \mathcal{M}} \left\langle \nabla_i c_i(x_{\mathcal{M}}^*, \bar{x}_{\mathcal{N} \setminus \mathcal{M}}), \bar{x}_i - x_i^t \right\rangle$$

$$= \eta_t \sum_{i \in \mathcal{N}} \left\langle \nabla_i c_i(x^*), x_i^* - x_i^t \right\rangle + \eta_t \sum_{i \in \mathcal{N}} \left\langle \nabla_i c_i(x_{\mathcal{M}}^*, \bar{x}_{\mathcal{N} \setminus \mathcal{M}}) - \nabla_i c_i(x^*), x_i^* - x_i^t \right\rangle$$

$$+ \eta_t \sum_{i \in \mathcal{N} \setminus \mathcal{M}} \left\langle \nabla_i c_i(x_{\mathcal{M}}^*, \bar{x}_{\mathcal{N} \setminus \mathcal{M}}), \bar{x}_i - x_i^* \right\rangle$$

$$\leq \eta_t \sum_{i \in \mathcal{N}} \ell_i \| x_i^* - x_i^t \| \left( \sum_{i \in \mathcal{N} \setminus \mathcal{M}} \| x_i^* - \bar{x}_i \| \right) + \eta_t \sum_{i \in \mathcal{N} \setminus \mathcal{M}} \left\| \nabla_i c_i(x_{\mathcal{M}}^*, \bar{x}_{\mathcal{N} \setminus \mathcal{M}}) \right\| \| \bar{x}_i - x_i^* \|$$

$$\leq O \left( \frac{\eta_t n B \sum_{i \in \mathcal{N}} \ell_i}{\sqrt{T}} + \frac{\eta_t n}{\sqrt{T}} \right).$$

Hence, as $D_p(x_i, x_i') \leq C_p, \forall x_i, x_i'$,

$$\sum_{i \in \mathcal{N}} D_p\left(x_i^*, x_i^{T+1}\right)$$

$$\leq O\left(\frac{n\nu \log(T)}{\eta_T \kappa T} + \frac{n\zeta B}{\eta_T T^{3/2}}\right) + O\left(\frac{nB\sum_{i \in \mathcal{N}} \ell_i}{\kappa T^{3/2}} + \frac{n}{\kappa T^{3/2}}\right) \frac{\sum_{t=1}^{T} \eta_t}{\eta_T} + O\left(\frac{nC_p}{\eta_T T}\right)$$

$$+ \frac{1}{\kappa\eta_T(T+1)} \sum_{i \in \mathcal{N}} \sum_{t=1}^{T} \eta_t \left\langle \hat{g}_i^t, x_i^t - x_i^{t+1}\right\rangle + \frac{1}{\kappa\eta_T(T+1)} \sum_{t=1}^{T} \eta_t \sum_{i \in \mathcal{M}} \left\langle \hat{g}_i^t - \nabla_i c_i\left(x^t\right), x_i^* - x_i^t\right\rangle$$

$$+ \frac{1}{\kappa\eta_T(T+1)} \sum_{t=1}^{T} \eta_t \sum_{i \in \mathcal{N} \setminus \mathcal{M}} \left\langle \hat{g}_i^t - \nabla_i c_i\left(x^t\right), \bar{x}_i - x_i^t\right\rangle .$$

$\square$

**Lemma J.2.** *Take $\eta_t \leq \frac{1}{2d}$, we have*

$$\left\langle \hat{g}_i^t, x_i^t - x_i^{t+1}\right\rangle = \eta_t \left\|A_i^t \hat{g}_i^t\right\|^2 .$$

*Proof.* Define

$$f(x_i) = \eta_t \left\langle x_i, \hat{g}_i^t\right\rangle + \eta_t(t+1)D_p(x_i, x_i^t) + D_h(x_i, x_i^t) .$$

As adding the linear term $\left\langle x_i, \hat{g}_i^t\right\rangle$ does not affect the self-concordant barrier property, and $p$ is strongly convex, $f(x)$ is a self-concordant barrier.

Define the local norm $\|h\|_x := \sqrt{h^\top \nabla^2 f(x)h}$, by Holder's inequality, we have

$$\left\langle \hat{g}_i^t, x_i^t - x_i^{t+1}\right\rangle = \left\|\hat{g}_i^t\right\|_{x_i^t, *} \left\|x_i^t - x_i^{t+1}\right\|_{x_i^t} .$$

Notice that

$$\nabla f(x_i^t) = \eta_t \hat{g}_i^t, \nabla^2 f(x_i^t) = \eta_t(t+1)\nabla^2 p(x_i^t) + \nabla^2 h(x_i^t) .$$

Therefore, by our assumption that $c_i(x) \in [0, 1]$,

$$\left\|\left(\nabla^2 f(x_i^t)\right)^{-1} \nabla f(x_i^t)\right\|_{x_i^t} = \eta_t \left\|A_i^t \hat{g}_i^t\right\|$$

$$\leq \eta_t d |c_i(\hat{x}^t)| \leq \eta_t d .$$

By Lemma J.4, take $\eta_t \leq \frac{1}{2d}$, we have

$$\left\|x_i^t - x_i^{t+1}\right\|_{x_i^t} = \left\|x_i^t - \arg\min_x f(x_i^t)\right\|_{x_i^t} \leq 2 \left\|\left(\nabla^2 f(x_i^t)\right)^{-1} \nabla f(x_i^t)\right\|_{x_i^t} \leq \eta_t \left\|A_i^t \hat{g}_i^t\right\| .$$

Therefore, we have

$$\left\langle \hat{g}_i^t, x_i^t - x_i^{t+1}\right\rangle = \eta_t \left\|A_i^t \hat{g}_i^t\right\|^2 .$$

$\square$

**Lemma J.3.** *[Proposition 1 Bauschke et al. (2017)] For an operator $G$ that $G - \nabla p(x)$ is monotone,*

$$\left\langle G(x) - G(x'), x' - x\right\rangle \leq -\sum_{i \in \mathcal{N}} \left(D_p\left(x_i, x_i'\right) + D_p\left(x_i', x_i\right)\right) .$$

*Proof.* By the monotonicity of $G - \nabla p(x)$, we have

$$\left\langle G(x) - G(x'), x' - x\right\rangle \leq \left\langle \nabla p(x) - \nabla p(x'), x' - x\right\rangle$$

$$\leq -\sum_{i \in \mathcal{N}} \left(D_p\left(x_i, x_i'\right) + D_p\left(x_i', x_i\right)\right) ,$$

where the second inequality is due to the definition of Bregman divergence.

$\square$

**Lemma J.4** (Lemma 3 Lin et al. (2021)). *For any self-concordant function $g$ and let $\lambda(x, g) \leq \frac{1}{2}$, $\lambda(x, g) := \|\nabla g(x)\|_{x,\star} = \left\|\left(\nabla^2 g(x)\right)^{-1} \nabla g(x)\right\|_x$, we have $\|x - \arg\min_{x' \in \mathcal{X}} g(x')\|_x \leq 2\lambda(x, g)$, where $\|\cdot\|_x$ is the local norm given by $\|h\|_x := \sqrt{h^\top \nabla^2 g(x) h}$.*

**Lemma J.5** (Lemma 7 of Lin et al. (2021)). *Suppose that $c_i$ is a convex function and $A_i \in \mathbb{R}^{d \times d}$ is an invertible matrix for each $i \in \mathcal{N}$, we define the smoothed version of $c_i$ with respect to $A_i$ by $\hat{c}_i(x) = \mathbb{E}_{w_i \sim \mathbb{B}^d} \mathbb{E}_{\mathbf{z}_{-i} \sim \Pi_{j \neq i} \mathbb{S}^d} [c_i(x_i + A_i w_i, \hat{x}_{-i})]$ where $\mathbb{S}^d$ is a d-dimensional unit sphere, $\mathbb{B}^d$ is a d-dimensional unit ball and $\hat{x}_i = x_i + A_i z_i$ for all $i \in \mathcal{N}$. Then, the following statements hold true:*

- $\nabla_i \hat{c}_i(x) = \mathbb{E}\left[d \cdot c_i(\hat{x}_i, \hat{x}_{-i})(A_i)^{-1} z_i \mid x_1, x_2, \dots, x_N\right]$.

- *If $\nabla c_i$ is $\ell_i$-Lipschitz continuous and we let $\sigma_{\max}(A)$ be the largest eigenvalue of $A$, we have $\|\nabla_i \hat{c}_i(x) - \nabla_i c_i(x)\| \leq \ell_i \sqrt{\sum_{j \in \mathcal{N}} (\sigma_{\max}(A_j))^2}$.*

**Lemma J.6** (Lemma 2 Lin et al. (2021)). *Suppose that $\mathcal{X}$ is a closed, convex and compact set, $R$ is a $\nu$-self-concordant barrier function for $\mathcal{X}$ and $\bar{x} = \arg\min_{x \in \mathcal{X}} R(x)$ is a center. Then, we have $R(x) - R(\bar{x}) \leq \nu \log\left(\frac{1}{1 - \pi_{\bar{x}}(x)}\right)$. For any $\epsilon \in (0, 1]$ and $x \in \mathcal{X}_\epsilon$, we have $\pi_{\bar{x}}(x) \leq \frac{1}{1+\epsilon}$ and $R(x) - R(\bar{x}) \leq \nu \log\left(1 + \frac{1}{\epsilon}\right)$.*

## K   MORE EXPERIMENTAL RESULTS

In Figure 2 and 3 we supplement more experiment results for zero-sum matrix games and Cournot competition. Note that in Figure 3, the curve of OMD with gradient coincides exactly with the curve GD with gradient. We found similar observations that our algorithm attains comparable performance to OMD and GD with full information gradient.

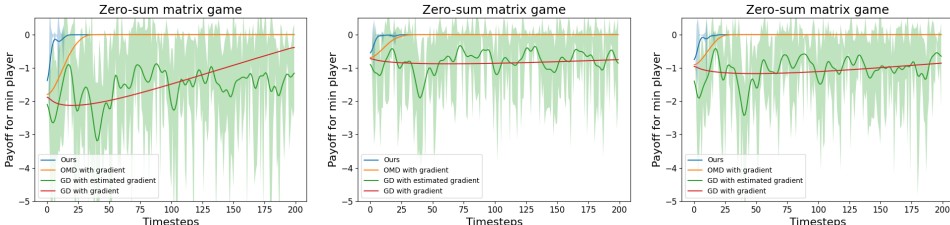

Figure 2: More examples on the zero-sum matrix game, with $A$ being $[[2, 1], [1, 3]]$, $[[3, 0], [0, 1]]$, and $[[1, 2], [2, 0]]$.

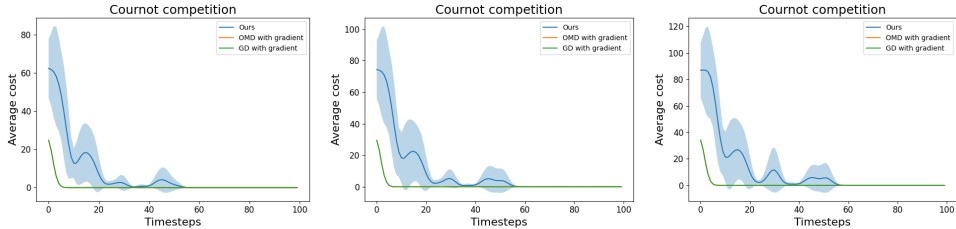

Figure 3: More examples on the Cournot competition, with the marginal cost being $50, 60, 70$.