# OpenReview forum: "Uncoupled and Convergent Learning in Monotone Games under Bandit Feedback"
_ICLR.cc/2025/Conference — ICLR 2025 Conference Withdrawn Submission_

### Official Review · Reviewer_ARbv · 2024-10-24

**Soundness:** 2
**Presentation:** 3
**Contribution:** 2
**Rating:** 5
**Confidence:** 4

**Summary:**

This paper studies the problem of learning in a general monotone game with bandit feedback, with strongly uncoupled dynamics. For smooth cost functions, the authors achieved an $O(T^{-1/4})$ last-iterate convergence rate. For strongly monotone games, the obtained last-iterate convergence matches the state-of-the-art rate $O(T^{-1/2})$. The proposed algorithm uses the standard gradient estimator for BCO, but updates with a slightly different OMD-based algorithm with a different regularizer.  When each player runs the same algorithm, the individual regret can also be guaranteed with $O(T^{3/4})$ regret for generally monotone games and $O(T^{1/2})$ for strongly monotone games. Finally, the authors further extended the results to time-varying games, considering two cases: (1) the time-varying game sequence gradually converges to a static game, and (2) the game sequence does not converge, and the performance measure relies on some non-stationarity measures depicting how the game sequence changes.

**Strengths:**

Extending the studies from strongly monotone games to generally monotone ones is meaningful. The proposed algorithm where a new regularizer is used is novel. The authors have also proved a high-probability result for the global convergence rate. And they have also applied their results to the social welfare to validate the importance of their results.

**Weaknesses:**

The core of achieving the last-iterate convergence is inserting a regularizer $p(\cdot)$ into the OMD update. Concretely, in the OMD update,  the authors imported an additional additive $\eta (t+1) D_p(x_i, x_i^t)$ into the OMD update. And the core analysis is shown in Lemma J.1 in the appendix. Specifically, by using $\sum_i D_p(x_i^*, x_i^{T+1})$ as the performance measure, the $\eta (t+1) D_p(x_i, x_i^t)$ in the OMD update can be extracted and moved to the left-hand side of the three-point equality of Bregman divergence, dividing both sides by $O(T)$ yields the performance measure on the left-hand side and a term like $(Regret)/T$ on the right-hand side. Consequently, it remains to analyze the regret guarantee of the proposed method and dividing it by $T$ yields the final last-iterate convergence rate.

This idea is novel. However, the key question is how we can access such function $p(\cdot)$. As stated by the authors on Page 5, the function $p(\cdot)$ has to satisfy the condition that $c_i(x_i, x_{-i}) - \kappa p(x_i)$ is convex. To compute such a valid $p(\cdot)$, the algorithm must have the complete information of the original cost function $c_i$, which is not permitted by the learning protocol since $c_i$ is unknown. Otherwise, if $p(\cdot)$ is given by the learning problem, the condition of $c_i(x_i, x_{-i}) - \kappa p(x_i)$ being convex should be an assumption that should be put in Section 3 along with Assumption 3.1. However, I am more inclined to believe that this should not be an assumption since $p(\cdot)$ is something that appears in the algorithm, while assumptions are actually statements about the problem-dependent and algorithm-independent quantities. Can the authors provide some further explanations on this issue?

**Questions:**

1. Can the $O(T^{-1/4})$ last-iterate convergence rate be considered as the $O(T^{3/4})$ regret for bandit convex optimization?

2. For bandit optimization with convex and smooth functions, the state-of-the-art regret is $O(T^{2/3})$ by [1], which seems to correspond to a last-iterate rate of $O(T^{-1/3})$ in the setup of smooth monotone games. Is it possible to improve the current $O(T^{-1/4})$ to $O(T^{-1/3})$?

3. For BCO problems with smooth or strongly convex functions, FTRL seems to be more popular than OMD [1,2]. Is it possible to represent Algorithm 1 using an FTRL-based update rule? Besides, are FTRL and OMD equivalent in the setup of BCO or bandit smooth monotone games?

4. The rates in Table 2 are not last-iterate results. First, the guarantees in Duvocelle et al. (2023) and Yan et al. (2023) are represented in a regret-type rate. Second, the rate in Theorem 6.2 is actually not last-iterate since the performance measure is $\frac{1}{T} \sum_t \cdots$. I suggest that the authors could correct this issue in the next version.

5. It seems that the rates authors mentioned in Duvocelle et al. (2023) and Yan et al. (2023) do not need smoothness? I did not check their results very carefully, and I suggest that the authors could conduct a careful double check about whether the above two works need smoothness. If not, these works seem to be not comparable since Duvocelle et al. (2023) and Yan et al. (2023) focused on strongly monotone games without smoothness while this paper considers monotone smooth games.

6. It should also be made clear that Duvocelle et al. (2023) and Yan et al. (2023) used tracking error $\\|x_t - x^*\\|^2$ as the performance measure, which is different from the measure used in this paper. Thus again I think these works are actually not comparable. Besides, if I am wrong about some statements, it is welcome that the authors could correct me.

6. In Line 287, the authors said that "We show that Algorithm 1 converges to the Nash equilibrium in monotone, strongly monotone, and linear games."  It should be made clear that the results for different kinds of games require different parameter configurations.

7. Could the authors provide some explanations about the validness of the permanence measure $\sum_i D_p(x_i^*, x_i^{T+1})$? Why this measure is a valid measure of the distance of the algorithm's output to the Nash equilibrium? The gap using Bregman divergence heavily relies on how convex the function $p(\cdot)$ is. If $p(\cdot)$ is nearly a linear function, this measure seems not to be a good enough performance measure, from my point of view, intuitively. I guess that the authors use this quantity as the performance measure mainly due to technical reasons, as I said in the 'Weaknesses' part?



References:

[1] Improved Regret Guarantees for Online Smooth Convex Optimization with Bandit Feedback, AISTATS 2011

[2] Bandit Convex Optimization: Towards Tight Bounds, NIPS. 2014

---

### Official Review · Reviewer_Tdey · 2024-10-27

**Soundness:** 2
**Presentation:** 2
**Contribution:** 2
**Rating:** 3
**Confidence:** 3

**Summary:**

This paper studies the convergence of mirror-descent algorithm in general monotone games and strongly monotone games. Several theoretical results are given. The time-varying monotone games are also considered. Overall I find the analyzed problem not new and the convergence rate is not surprised.

**Strengths:**

The paper provides a theoretical analysis on several cases, including the convergence rate for monotone games (both in expectation and with high probability), social welfare, linear cost, and the time-varying case.

**Weaknesses:**

- Many convergence rates in this paper fail to improve the existing result. The advantage of the proposed algorithm is unclear. The simulation results are also weak to demonstrate the advantage of the proposed algorithm.

- There is a large room for the simulation section to be improved. Please add comparison of Algorithm 1 to, at least, the method in [Lin 2021]. Besides, the examples used in simulation are simple.

- The technical contributions are unclear. The techniques used, including the ellipsoidal gradient estimator, are similar to those found in existing literature.

**Questions:**

- A big concern is the potential issue induced by the non-uniqueness of the Nash equilibrium for monotone games. In Theorem 5.1, convergence is measured by the distance of the actions to x_i^*. Does this statement hold for each x_i^*? Should the convergence be measured using the distance of the actions to the set of Nash equilibria?

- The concept of 'uncoupled' is quite confused. In games, each player's payoff depends on the actions of other agents, which suggests that their dynamics should be inherently coupled. However, a clear description of strongly uncoupled dynamics is not given.

- The assumptions required for each theorem are not explicitly stated, making it difficult for readers to follow directly from the theorem statements.

- The presentation of Algorithm 1 should be improved. For example, agents should first sample z and then play the perturbed action. Besides, each agent should receive feedback after all agents play their actions. Please see Algorithm 2 in [Lin 2021].

---

### Official Review · Reviewer_8rRA · 2024-11-03

**Soundness:** 3
**Presentation:** 3
**Contribution:** 3
**Rating:** 6
**Confidence:** 3

**Summary:**

The paper studies the last-iterate convergence of no-regret dynamics in monotone games under bandit feedback. In particular, they provide a new algorithm that attains the best-known rate of convergence of $T^{-1/4}$ and $T^{-1/2}$ under strong monotonicity, while guaranteeing at the same time the no-regret property. This is the first convergence rate for uncoupled dynamics in monotone games under bandit feedback. Further, for time-varying games, the convergence rate improves over the prior state-of-the-art.

**Strengths:**

The paper studies an important and well-studied problem in the intersection of optimization and game theory, and makes a number of concrete contributions that improve over the prior state-of-the-art. As described above, the paper obtains the best-known rates for monotone games and time-varying games under bandit feedback, which is certainly an important contribution. Further, the techniques employed in the paper are also quite non-trivial, and rely on using two different regularizers in an interesting way. While most prior results examine last-iterate convergence in the full-feedback model, the more realistic bandit model introduces several technical challenges. I believe that the results of the paper are well within the scope of ICLR, and will be well-received from the community on learning in games. The most related papers have been adequately discussed. All claims appear to be sound; I did not find any notable issue in the approach. The writing overall is of high quality.

**Weaknesses:**

In terms of weaknesses, it appears that the proof mostly relies on existing techniques, although the way those techniques are used seems to have new aspects. I would encourage the authors to highlight more the technical challenges and the technical contributions compared to prior work. There is also one issue that really confuses me: Section 5.3 is about the special case of linear cost functions; how is possible that the rate is worse in that special case? For example, matrix games are clearly monotone, so why doesn't the rate of $T^{-1/4}$ apply in that case?

**Questions:**

A couple of questions/issues for the authors:

1. In the third example, it is claimed that splittable routing games are monotone. Can the authors justify this claim? It is not obvious to me.
2. Before Proposition 5.1, the authors write that "We have the following proposition which shows that the social welfare converges to optimal welfare on average". This is not correct, the dynamics only approximate the optimal welfare (depending on the smoothness parameters).

---

### Official Review · Reviewer_WBMc · 2024-11-05

**Soundness:** 1
**Presentation:** 2
**Contribution:** 1
**Rating:** 3
**Confidence:** 3

**Summary:**

This paper design an algorithm for monotone games with bandit feedback that has last-iterate convergence guarantee.  The convergence rate is $T^{-1/4}$ for monotone games, and $T^{-1/6}$ for linear games.

**Strengths:**

- The authors attempted to improve the state-of-the-art last-iterate convergence rate for general monotone game and their time-varying variants --- the existing best bounds are $T^{-1/2}$ for strongly monotone games by Lin et al. (2021), and $T^{-1/6}$ for linear games by Cai et al. (2023).

**Weaknesses:**

- I have concern about the soundness of the results. At the first sight, there is a strange gap between the results for general monotone game and the linear game in Table 1:  for general monotone game the bound is $T^{-1/4}$, and for linear game the bound is $T^{-1/6}$.  However, linear game is a special case of monotone game. The author claimed that for linear games, one cannot find a convex function $p$ that satisfies the definitions in Line 217.  However, for any linear game, one can always artificially add a perturbation function $\epsilon ||x||^2$ to make it strictly monotone.  Applying the $T^{-1/4}$ bound for general monotone game, and bounding the error due to perturbation, one should be able to get a bound of $T^{-1/4}+\epsilon$ for linear games.  Then taking $\epsilon\to 0$, it seems we should also get a $T^{-1/4}$ bound also for linear games.

- It seems the main gap lies in Theorem 5.1.  In the theorem, although the right-hand side of the bound is $T^{-1/4}$, the left-hand side is not the usual Euclidean distance to the equilibrium, but the distance induced by the $p$ function defined in Line 217. This distance could be arbitrarily smaller than the Euclidean distance if $p$ is close to linear. Translating this distance back to Euclidean distance will just make this bound vacuous.  Therefore, I think the claim of the paper is flawed and it does not seem to make improvement over prior work.

**Questions:**

Please see the weakness section.

---

### Note · Authors · 2024-11-25

**Comment:**

We thank the reviewer for the constructive comments and we will revise the paper accordingly to prepare for a future venue.

**Withdrawal Confirmation:**

I have read and agree with the venue's withdrawal policy on behalf of myself and my co-authors.